# Visualization and standardized quantification of surface charge density for triboelectric materials

Yi Li [1,8], Yi Luo [2,8], Song Xiao[1,8], Cheng Zhang [2], Cheng Pan[1], Fuping Zeng[1], Zhaolun Cui[3], Bangdou Huang[2], Ju Tang[1], Tao Shao [2] ✉, Xiaoxing Zhang [4] ✉, Jiaqing Xiong [5] ✉ & Zhong Lin Wang [6,7] ✉

Triboelectric nanogenerator (TENG) operates on the principle of utilizing contact electrification and electrostatic induction. However, visualization and standardized quantification of surface charges for triboelectric materials remain challenging. Here, we report a surface charge visualization and standardized quantification method using electrostatic surface potential measured by Kevin probe and the iterative regularization strategy. Moreover, a tuning strategy on surface charge is demonstrated based on the corona discharge with a three-electrode design. The long-term stability and dissipation mechanisms of the injected negative or positive charges demonstrate high dependence on deep carrier traps in triboelectric materials. Typically, we achieved a 70-fold enhancement on the output voltage (~135.7 V) for the identical polytetrafluoroethylene (PTFE) based TENG (neg-PTFE/PTFE or posi-PTFE/PTFE triboelectric pair) with stable surface charge density (5% decay after 140 days). The charged PTFE was demonstrated as a robot e-skins for non-contact perception of object geometrics. This work provides valuable tools for surface charge visualization and quantification, giving a new strategy for a deeper understanding of contact electrification.

Contact-electrification (CE) as a universal but complex phenomenon has been known for over 2600 years[1]. However, its mechanism remains one of the oldest unsolved scientific puzzles due to the complex process[2]. The interfacial charge exchange/tunneling is a basic problem for comprehending the CE mechanism. With the invention and rapid development of triboelectric nanogenerator (TENG), tremendous efforts have been made to address this topic, significantly the charge transfer properties at interfaces[3–5]. So far, it remains challenging to characterize the surface charges by a direct visual. Hitherto, nanoscale investigations primarily utilized the Kelvin probe force microscopy (KPFM) based on the principle of atomic force microscopy (AFM)[6,7]. As for the macro scale, surface charge distributions could be visualized through dust figures by powdering with printer toner particles[8]. However, this will destroy the original charge distribution and cannot quantify the surface charge density. The Pockels effect technique could achieve online surface charge detection by utilizing the

[1]State Key Laboratory of Power Grid Environmental Protection, School of Electrical Engineering and Automation, Wuhan University, Wuhan, Hubei, People's Republic of China. [2]Beijing International S&T Cooperation Base for Plasma Science and Energy Conversion, Institute of Electrical Engineering, Chinese Academy of Sciences, Beijing, People's Republic of China. [3]School of Electric Power Engineering, South China University of Technology, Guangzhou, People's Republic of China. [4]Key Laboratory for High-Efficiency Utilization of Solar Energy and Operation Control of Energy Storage System, School of Electrical and Electronic Engineering, Hubei University of Technology, Wuhan, People's Republic of China. [5]Innovation Center for Textile Science and Technology, Donghua University, Shanghai, People's Republic of China. [6]Beijing Institute of Nanoenergy and Nanosystems, Chinese Academy of Sciences, Beijing, People's Republic of China. [7]School of Materials Science and Engineering, Georgia Institute of Technology, Atlanta, GA, USA. [8]These authors contributed equally: Yi Li, Yi Luo, Song Xiao. ✉e-mail: st@mail.iee.ac.cn; zhangxx@hbut.edu.cn; jqxiong@dhu.edu.cn; wangzhonglin@binn.cas.cn

relationship between electric field and light intensity, while the object should be transparent to ensure light penetration[9]. Advanced surface charge visualization and standardization methods are valuable tools for a deeper understanding of tribology, contact electrification, and topics related to TENG. For example, the correlation between macroscopic triboelectric performance and surface charge density could be elucidated. The evaluation of CE charge dynamic behavior (generation, storage, dissipation, etc) and diagnosis of surface defects affecting TENG performance can be accomplished by visualization and quantization of surface charge.

More recently, the electrostatic probe has been developed for surface potential detection in triboelectric materials[10–12]. The surface potential distribution generated by the triboelectric surface charge during TENG operation can be acquired by successively scanning the sample surface. Nevertheless, an inversion calculation procedure is required to determine the surface charge density from the surface potential based on their proportional relationship (Fig. 1a). Currently, a reversal calculation was performed to solve the surface charges from the measured potential using a constant transfer coefficient[7,13].

However, the surface potential at any position is determined by the distribution of all the surface charges, and the transfer coefficient changes with the probe position. To this end, it is necessary to develop a more precise computational approach that can effectively characterize the relationship between surface potential and surface charge, while also addressing the limitations of the transfer coefficient. Additionally, there is a pursuit for a charge imaging technique that offers high precision and computational efficiency to visualize the surface charge of triboelectric materials.

Besides, the superior output performance of TENG is an essential prerequisite of energy harvesting or self-powered sensing. Numerous methods have been proposed to boost the output performance, aiming to improve the effective surface charge density[14–17]. The high charge density facilitates exceptional triboelectric performance and holds potential for a wide range of applications, such as self-powered electronic skin (e-skin) for tactile sensing, biomechanical monitoring, human-machine interface, etc. The non-contact TENG that operates based on the principle of electrostatic induction could be achieved due to the exceptionally constant surface charge density[18]. One of the

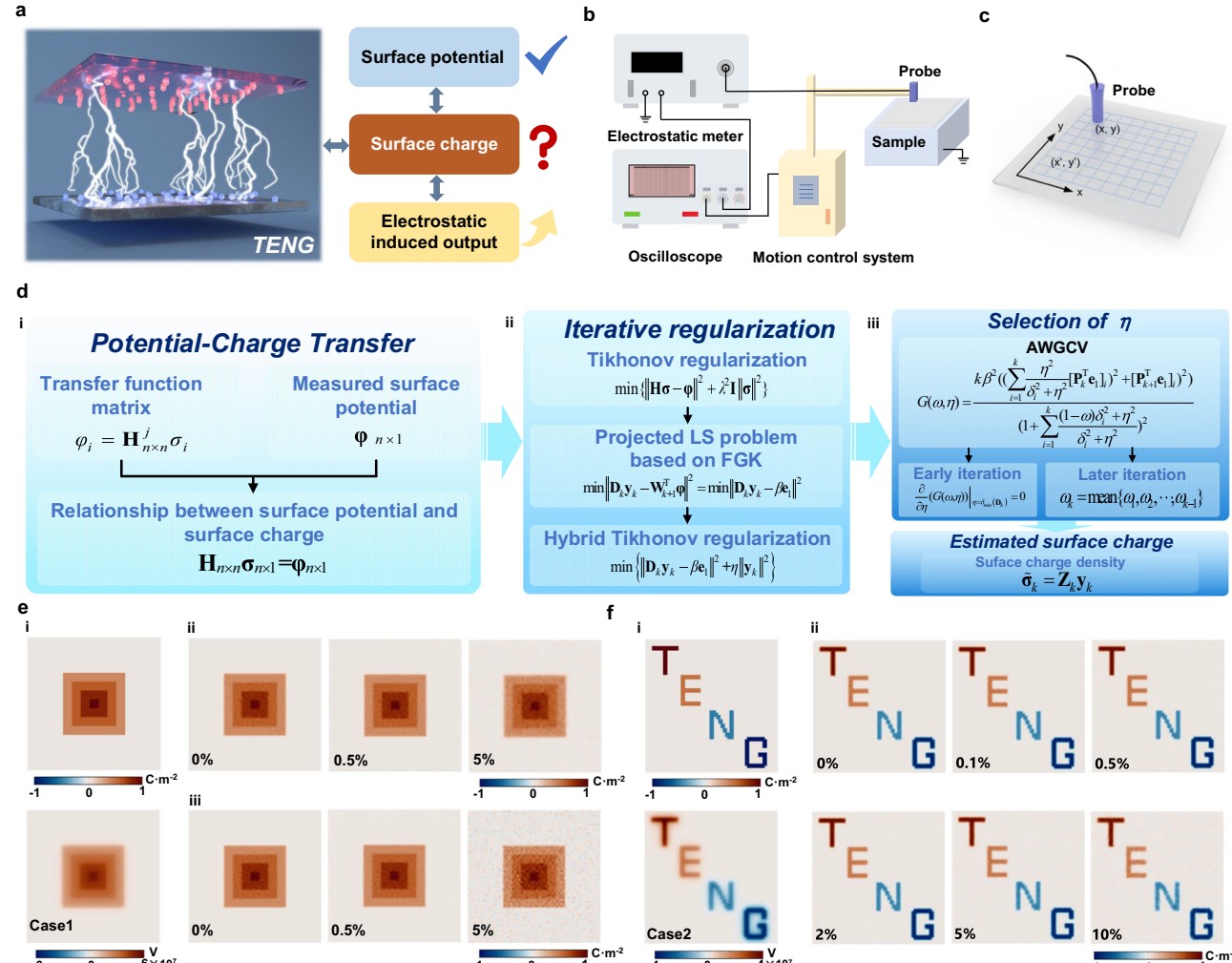

**Fig. 1 | Mechanism of the surface charge visualization and standardized quantification method. a** The correlation between surface potential and surface charge density for TENG. The contact-separation process of TENG generates triboelectric surface charge, which in turn creates a surface potential that impacts the electrostatic induction performance. **b** Schematic diagram of surface potential scanning platform. The waveform image originates from the data screenshot in the oscilloscope. **c** Schematic diagram of surface potential scanning trajectory. **d** Schematic illustration of the surface charge visualization and standardized quantification method. **e** Case 1 (Area range: 30 mm × 30 mm). (i) The artificially set "spotted-like" charges and the induced surface potential distribution. (ii) The obtained surface charge distribution under different Gaussian noise conditions by the VSQ method. (iii) The surface charge distribution under different Gaussian noises by the CS method. **f** Case 2 (Area range: 30 mm × 30 mm), (i) The artificially set "TENG" letter-like charges and the induced surface potential distribution. (ii) The obtained surface charge distribution under different Gaussian noise conditions by the VSQ method.

simple approaches to introduce charges into the triboelectric layer is through ion injection and corona discharge[19–21]. For example, Wang et al. utilized an air-ionization gun to inject single polarity ions (negative or positive) into the fluorinated ethylene propylene (FEP) film[20]. It is reported that the deposited positive charges have lower stability than negative charges. As for corona discharge injection, the ions generated by air ionization under high voltage were deposited on the sample surface[21,22]. The utilization of the tip-plane electrode produces a highly uneven electric field, leading to the creation of charge spots with an irregular distribution. Meanwhile, it is challenging to realize accurate surface charge deposition through corona discharge by traditional tip-to-plane electrodes. Regarding the selection of triboelectric material, the triboelectric series that describes the tendency of material capabilities to gain or lose electrons has been quantitatively standardized[23,24], wherein most of the polymers, inorganic non-metallic materials are prone to obtain electrons and the tribo-positive materials are scarce. Thus, realizing a tuning strategy on surface charge and controllable development of tribo-positive materials could effectively promote standardized characterization and diversified development of tribology and TENGs.

Herein, we achieved the visualization and standardized quantification of surface charges for triboelectric materials. The surface charge imaging method based on the flexible Golub-kahan hybrid approach was proposed first, which realizes the visualization and quantification of surface charge distribution using surface potential measured by the electrostatic probe. Further, we explored the interfacial charge exchange process of contact electrification and introduced a three-electrode system for single polarity charge injection. We achieved the triboelectric polarity tuning of various polymer films, such as polytetrafluoroethylene (PTFE), by injecting negative or positive charges. Additionally, the mechanisms of charge storage and dissipation, the long-term stability, and the involvement of charge carriers (shadow and deep traps) in triboelectric materials were confirmed. We showcased the identification of triboelectric layer defects by employing our proposed surface charge visualization technique. Besides, a robotic e-skin based on a posi-PTFE with high charge density was demonstrated for non-contact perception of object geometrics to highlight the tribo-polarity tuning effect. This work provides a beneficial tool for visualization and standardized quantification of surface charge, promising to advance a deeper understanding of contact electrification and customizable design of high-performance TENGs.

## Results

### Surface charge visualization and standardized quantification method

To obtain the surface potential distribution of triboelectric materials, we constructed a surface potential measurement platform based on active electrostatic probes. As shown in Fig. 1b, two stepper motors combined with programmable logic controllers were utilized to achieve high-precision movement control of the probe along $x$-axis and $y$-axis. The electrostatic probe undergoes an "S" shaped reciprocating motion during surface potential scanning (Fig. 1c and Supplementary Movie 1), and the real-time obtained charge information was stored based on the digital oscilloscope (Detailed information can be found in the "Methods" section). The surface potential ($\varphi$) distribution matrix ($60 \times 60$, a total of 3600 points) was obtained. Notably, the probe output corresponds to a linear superimposition of all the surface charges' effects on the sample surface. According to the Poisson's equation, the potential at any point in space can be expressed as,

$$\varphi(i) = \frac{1}{4\pi\varepsilon_0} \int_S \frac{\sigma}{r} dS \tag{1}$$

where $r$ represents the distance between point $i$ and any point on all surfaces $S$ in space.

Based on Eq. (1), the contribution of point $j$ to the potential of point $i$ can be written as,

$$\varphi(i) = \frac{1}{4\pi\varepsilon_0} \sum_{j=1}^{n} (\sigma)_j \int_{S_j} \frac{1}{r_{ij}} dS \tag{2}$$

where $S_j$ is the area of the $j^{\text{th}}$ element, $r_{ij}$ is the distance from point $i$ to point $j$, and the integral is on the local unit surface area.

Here, the measuring surface is divided into $n$ elements. It can be considered that the charges and potentials in the grid are evenly distributed when the grid is sufficiently small. Thus, the probe output ($\varphi(i)$) of the element $i$ can be given in discrete form,

$$\varphi(i) = \sum_{j=1}^{n} h_{ij} \cdot \sigma_j \tag{3}$$

where the coefficient $h_{ij}$ stands for the probe response at the point $i$ caused by a unit charge distributed at the element $j$. Further, the relationship between surface potential and charge density ($\sigma$) within each element can also be expressed in matrix form as (Fig. 1d(i)),

$$\begin{bmatrix} \varphi_1 \\ \vdots \\ \varphi_i \\ \vdots \\ \varphi_n \end{bmatrix}_{n \times 1} = \begin{bmatrix} h_{11} & \cdots & h_{1j} & \cdots & h_{1n} \\ \vdots & \ddots & \vdots & \ddots & \vdots \\ h_{i1} & \cdots & h_{ij} & \cdots & h_{in} \\ \vdots & \ddots & \vdots & \ddots & \vdots \\ h_{n1} & \cdots & h_{nj} & \cdots & h_{nn} \end{bmatrix}_{n \times n} \begin{bmatrix} \sigma_1 \\ \vdots \\ \sigma_i \\ \vdots \\ \sigma_n \end{bmatrix}_{n \times 1} \tag{4}$$

where $\varphi_i$ and $\sigma_i$ are the surface potential and surface charge density at point $i$, as shown in Supplementary Fig. 1. **H** is defined as the transfer function matrix between surface potential and charge. The surface charge density $\boldsymbol{\sigma}$ can be calculated by inversing the liner Eq. (4) once the components in **H** were collected. This process is commonly referred to as the charge simulation (CS) method. Notably, the matrix **H** is mostly ill-conditioned, whose sensitivity and stability for numerical calculation must be evaluated for the large linear systems (Detailed explanation was given in Supplementary Note 1). Thus, the solution of Eq. (4) can be regarded as a linear discrete ill-posed problem.

To solve Eq. (4), iterative regularization is considered to recover a meaningful approximation of the actual surface charge density $\boldsymbol{\sigma}_{\text{true}}$, as illustrated in Fig. 1d(ii). Theoretically, the ill-posed problem in Eq. (4) can be solved by standard Tikhonov regularization in a penalized least-squares form[25], namely, to define the regularized solution $\bar{\sigma}_\lambda$ as the minimizer of the following weighted combination of the residual norm and the side constraint,

$$\min\{\|\mathbf{H}\sigma - \varphi\|^2 + \lambda^2 \mathbf{I}\|\sigma\|^2\} \tag{5}$$

Equation (5) is equivalent to the regularized linear equations,

$$(\mathbf{H}^{\mathsf{T}}\mathbf{H} + \lambda^2 \mathbf{I})\bar{\sigma}_\lambda = \mathbf{H}^{\mathsf{T}}\varphi \tag{6}$$

where **I** is the identity matrix, $\lambda$ ($\tau_n < \lambda < \tau_1$) is the regularization parameter, $\tau_1$ is the minimum singular value of the matrix **H**, and $\tau_n$ is the maximum singular value of the matrix **H**. In general, the estimated surface charge distribution $\bar{\sigma}_\lambda$ can be well obtained by inversing Eq. (6) with a reasonable regularization parameter $\lambda$. However, the large number of measurement points usually results in a vast dimension of the transfer function matrix where the standard Tikhonov method is infeasible concerning computing time or storage. Moreover, although obtaining the cut-off frequency based on the point spread function to determine the regularization parameter can ensure a certain computation accuracy, the predefined regularization parameters cannot

achieve the best results when managing measurement data with different noise levels. In this regard, we propose a visualization and standard quantitation (VSQ) method to achieve high accuracy in the presentation of surface charge.

Accordingly, we introduced a Flexible Golub-Kahan (FGK) method to solve the ill-posed problem. Given the FGK decomposition as discussed in Supplementary Note 2, the least squares problem (5) can be transformed into the following projected squares problem when considering the data-fit term,

$$\min\left\{\left\|\mathbf{D}_k\mathbf{y}_k - \beta\mathbf{e}_1\right\|^2\right\} \tag{7}$$

where, $\mathbf{y}_k$ is the solution of the ill-posed problem (7), and $\tilde{\sigma}_k = \mathbf{Z}_k\mathbf{y}_k$, $\mathbf{Z}$ is the column space to span the subspace of FGK decomposition, as shown in Supplementary Note 2, $\mathbf{e}_1$ is the first column of the identity matrix of order $k+1$. However, the so-called "semi-convergence" phenomenon occurs directly in solving Eq. (7), where early iterations reconstruct information about the solution while later iterations reconstruct information about the noise[26]. To this end, the projected least squares problem (7) can be solved by the form of Tikhonov regularization,

$$\min\left\{\left\|\mathbf{D}_k\mathbf{y}_k - \beta\mathbf{e}_1\right\|^2 + \eta\left\|\mathbf{y}_k\right\|^2\right\} \tag{8}$$

where $\eta$ is the regularization parameter of Eq. (8).

The choice of the regularization parameter has an essential influence on the accuracy of the image calculation and directly affects the similarity between the approximate and real solutions. The standard generalized cross validation (GCV) is one of the most representative approaches to estimating the regularization parameter, which is based on the philosophy that if an arbitrary element $\varphi_i$ in potential matrix $\boldsymbol{\varphi}$ is left out, then the corresponding estimated solution should predict the observation well (as discussed in Supplementary Note 3). However, the GCV function usually causes overestimation with the large regulation parameter selected for the projected least squares problem (7)[27]. Accordingly, we proposed the adaptive weight generalized cross validation (AWGCV) to select the regularization parameter $\eta$, as illustrated in Fig. 1c(iii)[28,29]. The WGCV function of the projected least squares problem (8) is given as follows,

$$G(\omega,\eta) = \frac{k\left\|(\mathbf{I} - \mathbf{D}_k\mathbf{D}_{k,\eta}^{\dagger})\beta\mathbf{e}_1\right\|_2}{\left(trace(\mathbf{I} - \omega\mathbf{D}_k\mathbf{D}_{k,\eta}^{\dagger})\right)^2} \tag{9}$$

Perform the singular value decomposition (SVD) of matrix $\mathbf{D}_k$,

$$\mathbf{D}_k = \mathbf{P}_{(k+1)\times(k+1)}\begin{bmatrix}\Delta_{k\times k}\\\mathbf{0}^{\mathrm{T}}\end{bmatrix}\mathbf{Q}_{k\times k}^* \tag{10}$$

where $\mathbf{P}$ is a unitary matrix of order $(k+1)\times(k+1)$; $\boldsymbol{\Delta} = \mathrm{diag}(\delta_1, \delta_2, ..., \delta_k)$ is a diagonal matrix of order $k\times k$, and the elements of its diagonal are the singular values of the matrix $\mathbf{D}_k$ that arranged in the order of $\delta_1 \geq \delta_2 \geq \cdots \geq \delta_k \geq 0$; $\mathbf{Q}^*$ is the conjugate transpose of $\mathbf{Q}$, which is a unitary matrix of order $k\times k$. The regularization solution of (8) is equivalent to[29]:

$$\mathbf{y}_k = \sum_{i=1}^{k}\psi_i\frac{p_i^{\mathrm{T}}\beta\mathbf{e}_1}{\delta_i}q_i \tag{11}$$

where, $p_i$ and $q_i$ are the elements in the matrices $\mathbf{P}$ and $\mathbf{Q}$, respectively, $\psi_i = \frac{\delta_i^2}{\delta_i^2 + \eta^2}$ represents the Tikhonov filter factor with its range in interval $[0, 1]$.

Using the relation of (9) and (11), the WGCV function is transformed into:

$$G(\omega,\eta) = \frac{k\beta^2\left(\left(\sum_{i=1}^{k}\frac{\eta^2}{\delta_i^2 + \eta^2}\left[\mathbf{P}_k^{\mathrm{T}}\mathbf{e}_1\right]_i\right)^2 + \left[\mathbf{P}_{k+1}^{\mathrm{T}}\mathbf{e}_1\right]_i^2\right)}{\left(1 + \sum_{i=1}^{k}\frac{(1-\omega)\delta_i^2 + \eta^2}{\delta_i^2 + \eta^2}\right)^2} \tag{12}$$

Define that $\eta_{k,\,\mathrm{opt}}$ is the optimal regularization parameter of the $k$th iteration. The regularization parameter has little effect in early iterations because the gap between the regularized solution and the actual value is large. The $\eta_{k,\,\mathrm{opt}}$ can be assumed as:

$$0 \leq \eta_{k,opt} \leq \delta_{\min}(\mathbf{D}_k) \tag{13}$$

where $\delta_{\min}(\mathbf{D}_k)$ is the minimum singular value of matrix $\mathbf{D}_k$ in each iteration. Assuming that $\eta_{k,\,\mathrm{opt}}$ is known, the $\omega$ can be found by minimizing the GCV function by the partial derivative with respect to $\eta$ from (12), namely,

$$\frac{\partial}{\partial\eta}(G(\omega,\eta))\Big|_{\eta=\eta_{k,opt}} = 0 \tag{14}$$

Since $\eta_{k,\,\mathrm{opt}}$ is unknown, we instead find $\omega$ corresponding to $\eta_{k,opt}=\delta_{\min}(\mathbf{D}_k)$. In later iterations, this approach fails because $\delta_{\min}(\mathbf{D}_k)$ becomes nearly zero due to ill-conditioning. Thus, in order to prevent $\omega$ from being too small in subsequent iterations, the $\omega_k$ at $k$th iterations is the average value in the previous iterations, namely,

$$\omega_k = \mathrm{mean}\{\omega_1,\omega_2,\cdots,\omega_{k-1}\} \tag{15}$$

Besides, a reasonable iterative stop criteria is required to terminate the iteration to prevent under-smoothing when $k$ is too large. Ideally, the convergence process of the hybrid method is perfectly stabilized. Therefore, the iteration can be terminated when the tolerance condition ($tol$) is satisfied:

$$\left|\frac{G(\omega,\eta)_{k+1} - G(\omega,\eta)_k}{G(\omega,\eta)_1}\right| < tol \tag{16}$$

where $tol$ is generally chooses $10^{-12}$. Given the fact that the semi-convergent behavior of the iterations cannot be completely circumvented, the iteration is also stopped when $G(\omega,\eta)$ starts to increase within a certain of steps. At which time, the iteration stopping step $k$ returns the corresponding value with the global minimum of $G(\omega,\eta)$[29],

$$k_{stop} = \mathrm{argmin}_k(G(\omega,\eta)) \tag{17}$$

To verify the performance of the proposed VSQ method, we artificially set up two forms of surface charge. In Case 1, a "spotted-like" charge distribution with a maximum value of $1\,\mathrm{C\,m^{-2}}$ emanating from the center is established (Fig. 1e). In Case 2, we set a letter pattern of "TENG" (Fig. 1f) and the corresponding charge density is assigned to $1\,\mathrm{C\,m^{-2}}$, $0.5\,\mathrm{C\,m^{-2}}$, $-0.5\,\mathrm{C\,m^{-2}}$ and $1\,\mathrm{C\,m^{-2}}$, respectively. The surface potential distribution induced by the established charge is calculated based on the Poisson equation. Besides, the Gaussian noise, whose standard deviation equals 0.1%, 0.5%, 2%, 5%, and 10% of the maximum surface potential $\boldsymbol{\varphi}$, is superimposed to simulate the interference signal in the actual surface potential test. Accordingly, we utilized the proposed VSQ to estimate charge distribution from the obtained surface potential. As shown in Fig. 1e(ii) and Supplementary Fig. 2a, the estimated results are highly consistent with the set surface charge pattern (Case 1). The background noise is nicely inhibited even when it reaches up to 10%. Analogously, this method can also restore the

charge distribution in Case 2 under different noise levels, as shown in Fig. 1f(ii). To further validate the stability and precision of the proposed VSQ method, we employed the CS method to compute the charge distribution of Case 1 (Fig. 1e(iii) and Supplementary Fig. 2b) and Case 2 (Supplementary Fig. 3) under identical noise conditions. In subjective image observation, both the VSQ and the CS methods yield superior visual outcomes when the noise level is less than 2%. However, when the noise level exceeds 2%, the charge distribution obtained by the CS method becomes indistinguishable due to the presence of noise.

Further, we present the relative error, signal-to-noise ratio (SNR), and peak mean square error (PMSE) as objective measures to assess the accuracy and image quality. The detailed methodology for these evaluations is described in Supplementary Note 4. Figure 2a illustrates the relative error of the VSQ and CS methods for the two cases, where

the relative error of charge inversion results exhibits a progressive increase as the noise levels rise. The conventional CS method is unable to produce satisfactory outcomes when the noise is above 2%. Specifically, the relative errors from the CS method achieved 41.40% (Case 1) and 60.70% (Case 2) when the noise reached 10%. In contrast, our proposed VSQ algorithm achieved significantly lower relative errors of 13.36% (Case 1) and 15.46% (Case 2), respectively. The SNR and PMSE are crucial metrics for assessing the image quality, and the corresponding outcomes are shown in Supplementary Fig. 4 and Fig. 2b. Regarding Case 1 and Case 2, the SNR obtained by the CS method is apparently lower than that of the VSQ method, and the PMSE is higher. Furthermore, as the level of noise rises, the ill-posed problem of the linear system (4) becomes more pronounced, resulting in a more noticeable discrepancy. In summary, the comparison of the relative error, SNR, and PMSE confirms that our suggested visualization and

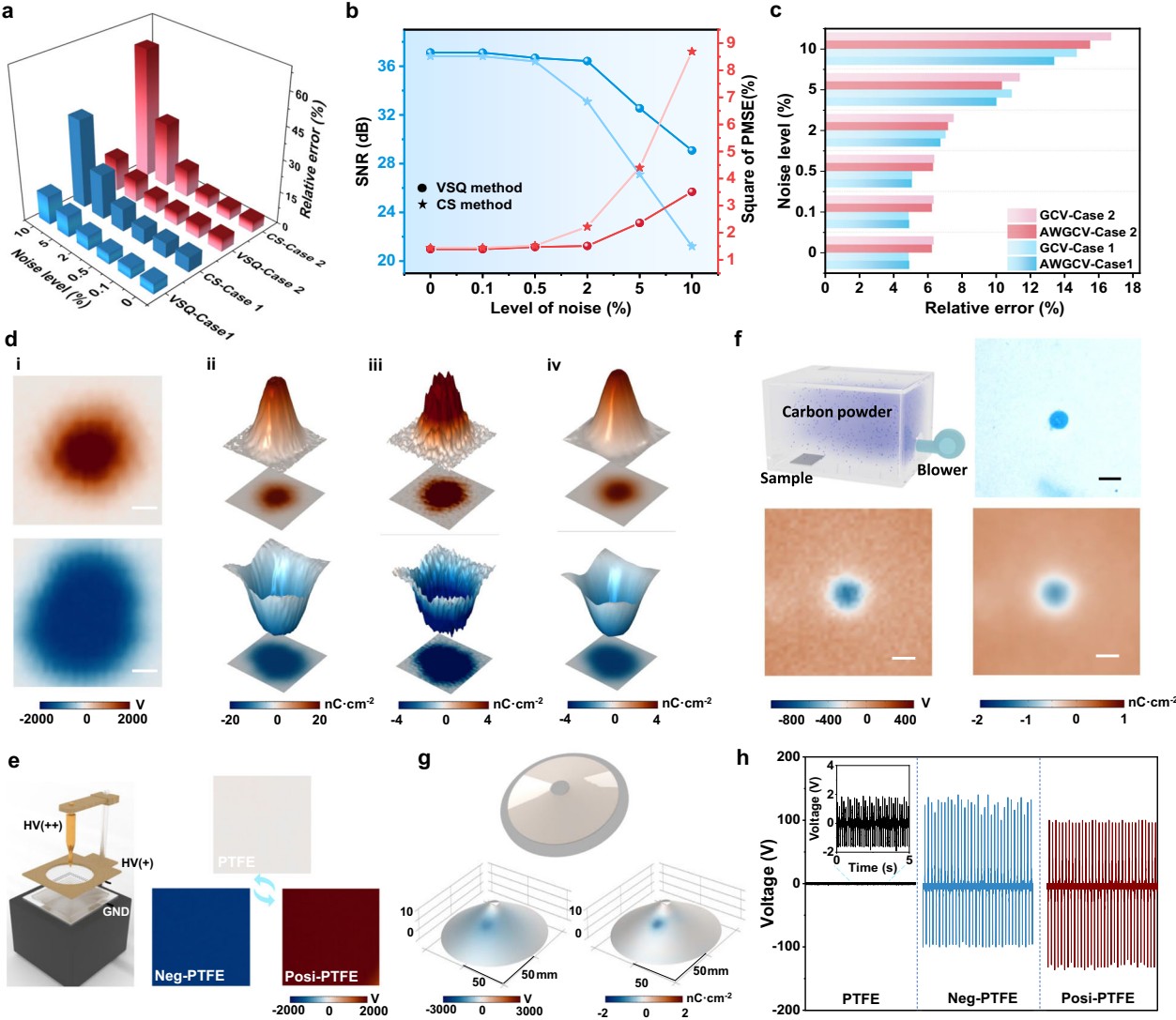

Fig. 2 | Accuracy assessment of the surface charge visualization and standardized quantification method. a Comparison of the relative error of the VSQ and CS methods regarding Case 1 and Case 2 under different noise levels. b Comparison of the SNR and PMSE of the VSQ and CS methods regarding Case 2 under different noise levels. c Comparison of the relative error of the AWGCV and GCV approach based on the VSQ method regarding Case 1 and Case 2. d (i) The surface potential distribution of the charge spot induced by the tip-plane electrode. (ii) the surface charge density obtained by the capacitor model. (iii) the surface charge density obtained by the CS method. (iv) the surface charge density obtained by the VSQ method (Scale bar: 5 mm). e Schematic illustration of the three-electrode for

corona discharge induction and the triboelectric charge polarity tuning of PTFE. The HV(++) and HV(+) indicates the high voltage applied to the tip and metallic grid electrodes. The right images represent the PTFE film with positive, negative, and non charges. f Comparison of the surface charge distribution obtained by dust figure (powdering with printer toner particle) and the VSQ method (Scale bar: 1 cm). g Demonstration of the VSQ method for surface charge imaging of three-dimensional structure sample. h The output performance of PTFE/PTFE, neg-PTFE/PTFE, and posi-PTFE/PTFE triboelectric pairs based TENGs. The inset demonstrates the partially enlarged image of PTFE/PTFE output.

standardized quantification method possess exceptional resistance to interference and high accuracy in inversion.

In addition, to further illustrate the necessity and accuracy of the AWGCV, we also applied the GCV to calculate charge inversion results for Case 1 and 2 based on the VSQ method, and the obtained surface charge distributions are shown in Supplementary Fig. 5. According to Supplementary Fig. 6, Case 1 and 2 both exhibit considerable inversion capability with GCV approach under low noise conditions. As the level of noise rises, the SNR of the GCV method decreases in comparison to the AWGCV method, and the PMSR increases. Furthermore, the GCV approach presents an over-smoothing solution for Case 1 and 2 at high noise levels (5% and 10%), resulting in a maximum charge value that is less than $1\,C\,m^{-2}$. The relative error exhibits a similar pattern, whereby the accuracy of the GCV method is inferior to that of the AWGCV method when subjected to high levels of noise (Fig. 2c).

Algorithm performance validation using measurable data has been consistently conducted. Here, we first created a charge spot on the PTFE surface and compared the imaging performance of CS and VSQ methods. As shown in Fig. 2d, the presence of noise has significantly compromised the accuracy of the charge produced by the CS approach, rendering it indistinguishable and obscuring the obvious identification of the charge's edge. We also deposited evenly distributed negative or positive charges into the PTFE film based on a three-electrode system (Fig. 2e) induced corona discharge. As shown in Supplementary Fig. 7, the deposited charges generated a positive or negative surface potential of 2017 V, and −1705 V, respectively. The charge magnitude determined by the VSQ method is equivalent to that of the CS method. Besides, the surface charge density evaluated by the traditional capacitor model ($\sigma = \varepsilon \frac{\varphi}{d}$, where $\varepsilon$ and $d$ represents the dielectric constant and sample thickness[7]) is about 4 times higher than that of the VSQ and CS method. Notably, the capacitor model assumes that the observed site's surface potential is proportional to its surface charge density, which means that the surface charge distribution is determined using a linear scale approach. The potential of a measured site is produced by the superposition of the local surface charges and their surroundings. Therefore, the conventional capacitor model is inadequate for charge inversion. To further verify the accuracy of the VSQ method through experimental means, we utilized the "dust figure" method to visually demonstrate the surface charge distribution. The process for creating the dust figure is outlined in Supplementary Note 5. Figure 2f shows the charge distribution obtained by the VSQ method is consistent well with the actual distribution measurement results based on the dust figure. Both experimental and simulation results indicate that this VSQ method approach exhibits exceptional resistance to interference, along with greater precision in quantifying surface charge.

The versatility and universality of the VSQ method have also been verified. The **H** matrix is independent of the distance between the electrostatic probe and the sample according to Eqs. (2) and (3). Instead, it simply depends on the distance between points $i$ and $j$, as well as the shape of the tested sample and the positioning of the ground electrode. The change in probe-to-surface distance will only impact the precision of surface potential measurement, and there are no demands for the device or potential measurement parameter settings (as discussed in Supplementary Note 6 and Supplementary Fig. 8). Therefore, the method suggested in this study can also be employed to solve the related **H** matrix to achieve charge inversion based on specific structures. As a proof of concept, we exemplified the utilization of this approach in the context of three-dimensional (3D) surface charge reversal. Specifically, we created a truncated cone-shaped sample and employed a circular trajectory for scanning its surface distribution. We successfully visualized the surface charge distribution using the VSQ method, as depicted in Fig. 2g.

Besides, it was unexpected to discover that the surface charge polarity of PTFE can be tuned, despite its common usage as a tribo-negative material. We further explored the triboelectric performance of PTFE, neg-PTFE ($-3.02\,nC\,cm^{-2}$) and posi-PTFE ($3.54\,nC\,cm^{-2}$) with a couple material of PTFE film, as shown in Fig. 2h. The PTFE-PTFE couple produces an output voltage of 1.84 V, which is ascribed to the micro/minor curved surface changes the surface state energy levels and causes an electron transition during CE (Detailed explanation was given in Supplementary Note 7 and Supplementary Figs. 9 and 10)[30,31]. In comparison, the output voltage of 135.72 V was achieved for the couple of neg-PTFE/PTFE, and 135.59 V with reverse polarity waveform can be produced by the posi-PTFE/PTFE couple, both of them show ~70 fold enhancement on the output. Therefore, unlike traditional high-performance TENGs that rely on materials with large differences in charge affinities, we first verified that either negative or positive charges can be realized on an identical material, with tunable surface charge densities by controlling injection parameters to achieve superior output. The comprehensive explorations will be deliberated in the subsequent sections.

## Surface charge tuning for triboelectric materials

The origin of surface charges is one of the most fundamental subjects for TENG. So far, there is a deficiency in the ability to visualize and establish a consistent method for quantifying the creation and dissipation processes of triboelectric charges that occur through CE. Here, utilizing the VSQ method, we systematically explored the evolution of surface charges based on a contact separation TENG with PTFE-Aluminun triboelectric pair (the residual charge on the PTFE surface was removed by air-ionization gun, Supplementary Fig. 11) and Aluminum electrode. As shown in Fig. 3a and Supplementary Fig. 12, the output voltage of the TENG increased from 28.24 V to 67.36 V as the contact number changed from 1500 to 9000. Meanwhile, the surface potential of PTFE also increased from −102.39 V (1500 times) to −299.81 V (9000 times).

The transferred charge ($Q_{sc}$) and surface charge (electrostatic charge) represent similar properties, as shown in Supplementary Figs. 13 and 14 and Fig. 3c. Specifically, the negative electrostatic charge was transferred from Aluminum to PTFE due to the negative triboelectric polarity of PTFE. The surface charge density of PTFE initially increased from $3.08 \times 10^{-4}\,nC\,cm^{-2}$ (before contact) to $-0.46\,nC\,cm^{-2}$ after 6000 times contact, then gradually rose to $-0.61\,nC\,cm^{-2}$ (9000 times). Thus, the amount of surface charge is directly linked to the duration of contact-separation periods, which is consistent with the KPFM mechanism[32]. Meanwhile, the surface charge density trends to reach saturation after 7500 instances of contact, suggesting that the net charges produced by contact electrification are restricted. Generally, the surface potential produced by the electrostatic charges results in a redistribution of free charges in the electrodes, and the $Q_{sc}$ belongs to the coupling of electrostatic and free charges. Obviously, the value of surface charge generated under different contact times accounts for ~23% of the transferred charge, indicating that free charge contributes dominance for TENG. Therefore, the improvement of surface electrostatic charge will also contribute to the induction of free charge, boosting the output performance of TENG.

Surface charge tuning through charge injection or deposition by corona discharge has been widely adopted for boosting the triboelectric performance of TENG[19–21]. However, the visualization and quantification of injected surface charges cannot be estimated exactly in advance[33]. Meanwhile, the commonly utilized tip-plane electrode for corona discharge generates charge spots on the surface (Supplementary Fig. 15), which is ascribed to the air collision ionization induced by the non-uniform electric field concentrated at the tip. To improve the uniformity of the induced surface charge, we proposed a three-electrode system composed of the tip, metallic grid, and plane electrode. As illustrated in Fig. 3d, the high negative voltage (-8 kV) applied to the tip will induce air ionization (corona discharge), where the

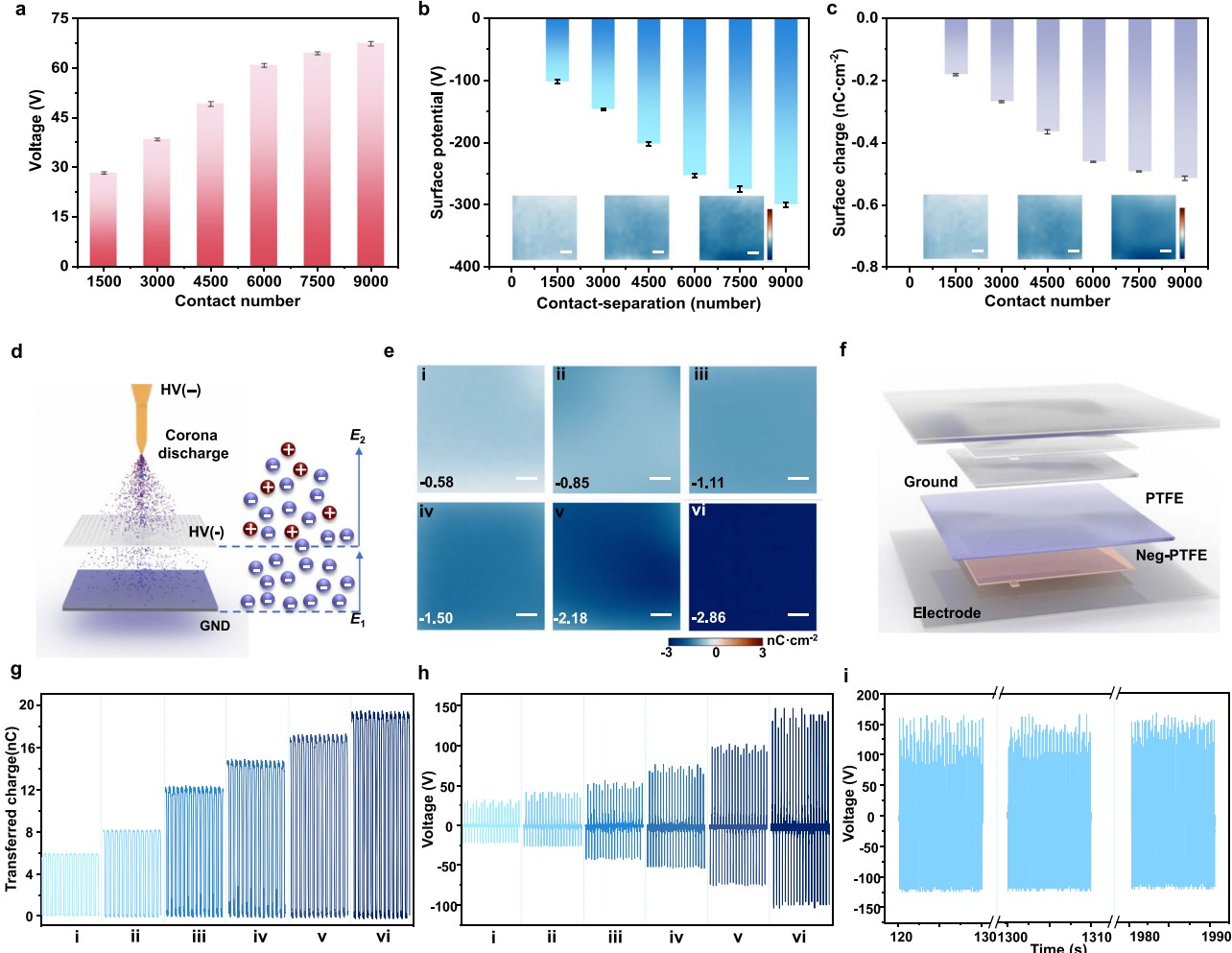

**Fig. 3 | Negative surface charge tuning and standardization for triboelectric materials. a–c** The (**a**) output voltage (same sample measured repeatedly, mean ± s.d., $n = 5$), (**b**) surface potential (same sample measured repeatedly, mean ± s.d., $n = 5$. The inset images demonstrate the surface potential of PTFE after 1500, 4500, and 7500 contact-separation, scale bar: 3 mm, color bar: −600 V - 600 V), (**c**) surface charge density (same sample measured repeatedly, mean ± s.d., $n = 5$. The inset images demonstrate the surface charge density of PTFE after 1500, 4500, and 7500 contact-separation, scale bar: 3 mm, color bar: −1 nC cm⁻² - 1 nC cm⁻²) development trend of PTFE-based TENG within 9000 times of contact with Aluminum ($2 \times 2$ cm²,

$F = 50$ N). **d** Schematic illustrations of the three-electrode system for single polarity charge injection. **e** Tuning and standardization of negative surface charge on PTFE, (i) −0.58 nC cm⁻², (ii) −0.85 nC cm⁻² and (vi) −2.86 nC cm⁻² (Scale bar: 3 mm). **f** Schematic illustrations of the neg-PTFE/PTFE triboelectric pair-based TENG. **g, h** The (**g**) transferred charge ($Q_{sc}$), (**h**) output voltage of neg-PTFE/PTFE triboelectric pair based TENG with different negative surface charge density ($2 \times 2$ cm², $F = 50$ N). **i.** The long-term output stability of neg-PTFE/PTFE triboelectric pair-based TENG ($2 \times 2$ cm², $F = 50$ N).

negative ions and electrons move to the grounded plane electrode and the positive charges migrate to the tip electrode. A lower negative voltage (-2 kV) was given to the grid electrode to neutralize positive charges and provide a channel for electrons and negative ions to the sample surface, ensuring the even injection of surface charges. Using this method, PTFE films with different values of uniform negative surface charge densities were obtained by adjusting the corona discharge duration time (Detailed information was given in Supplementary Note 8 and Supplementary Figs. 16 and 17). As shown in Fig. 3e and Supplementary Fig. 18, the maximum surface potential and surface charge density of -1618 V and -2.86 nC cm⁻² were achieved, respectively. Subsequently, the triboelectric performance related to the injection density of surface charge was studied, using the pair of neg-PTFE/PTFE with TENG structure as depicted in Fig. 3f. As expected, the injection of surface charge density effectively boosted the output performance. Compared to the PTFE-PTFE pair delivered an output voltage of 1.84 V, the neg-PTFE with the surface charge density of -0.58 nC cm⁻², −0.85 nC cm⁻², −1.11 nC cm⁻², −1.50 nC cm⁻², −2.18 nC cm⁻², and −2.86 nC cm⁻² (Fig. 3e) showed the transferred

charge increased from 5.88 nC to 17.25 nC (Fig. 3g), which produce a voltage of 31.65 V, 41.75 V, 53.15 V, 75.31 V, 102.63 V, 135.72 V, respectively (Fig. 3h). To the best of our knowledge, this is the highest output achieved by the TENG consisting of identical material. The long-term output performance of the device with high stability confirms the injected surface charge was preserved (Fig. 3i). The surface charges of both neg-PTFE and PTFE after frequent testing were monitored, as shown in Supplementary Figs. 19 and 20, which revealed that the surface charge density of neg-PTFE reduced by 10−20% while the uncharged PTFE gained a surface charge of −0.07 nC cm⁻² to −0.49 nC cm⁻² (Supplementary Fig. 21). This is attributed to the surface charge transfer induced by the potential difference between neg-PTFE and PTFE. The neg-PTFE with higher surface charge density delivers much more charge during operation, while the mutual repulsion effect between the negative charges partially hinders the interfacial charge transport, resulting in relatively stable output performance.

So far, there are few materials with high positive polarity in the triboelectric series[34,35], which are important for coupling with negative

materials to realizing high-performance TENG[36]. Although positive charge deposition by air-ionization guns has been explored, their stability is substantially worse than that of injected negative charges[20]. Here, we realized stable deposition and tuning of positive charge on PTFE by the three-electrode induced corona discharge. Figure 4a–c and Supplementary Fig. 22 present posi-PTFE with a surface charge density of 0.5–3.56 nC cm⁻². The output voltage and transferred charge based on the single electrode mode were also measured to confirm the improved effect of triboelectric performance. Typically, the posi-PTFE with a charge density of 3.56 nC cm⁻² produced an output voltage of 135.59 V and $Q_{sc}$ of 18.01 nC, which is 6.05 and 3.59 times for the surface with 0.5 nC cm⁻² charge density, respectively. Long-term stability of the posi-PTFE-based TENG was verified even after 1500 s of operation (Supplementary Fig. 23). Moreover, interfacial transfer of the positive charges also occurred (Supplementary Figs. 24–26). For example, the surface charge density of the posi-PTFE reduced to 2.98 nC cm⁻², while the uncharged PTFE obtained 0.53 nC cm⁻². Thus, PTFE can be utilized as either negative or positive triboelectric materials for high-performance TENG.

Thereafter, we investigated the polarity tuning and surface charge deposition behavior of various common triboelectric materials, including Perfluoroalkyl (PFA), Polyvinyl chloride (PVC), Polyethylene terephthalate (PET), Polypropylene (PP), Polyethylene (PE), Silicon rubber (Si rubber), Polyamide (Nylon) and glass. The positive polarity corona discharge was employed to inject charges and establish different surface charge densities in this case. As shown in Fig. 4d,

positive charges can be injected to most polymers with a density of 3.41–3.56 nC cm⁻². However, Nylon and glass exhibit lower positive charge acceptances with the surface charge density of 0.25 nC cm⁻² and 0.02 nC cm⁻², respectively (the reason is discussed in Section 2.3). Meanwhile, the triboelectric performance of the above post-materials was further explored by contacting with the same uncharged film. Figure 4e and Supplementary Fig. 27 show that the initial output voltage of posi-PTFE, posi-PFA, posi-PET attained 130–140 V, followed by the posi-Si rubber (86 V), and posi-PVC (82 V). However, the posi-PP and posi-PE produced initial output voltages of 69.3 V and 40 V, respectively. It is reasonable that PFA, PVC, PET, and Si rubber are typical materials with negative tribo-polarity, PE, PP, Nylon, and glass belong to tribo-positive materials[23]. Here, we validate that surface charge injection can effectively modify the tribo-polarity of different triboelectric materials, allowing a transition from negative to positive.

In addition, there exhibits apparent output voltage attenuation for the posi-PE and posi-Si rubber (Supplementary Fig. 27 and Fig. 4e), suggesting the severe interfacial charge transport between the positively charged and uncharged surfaces. As verified by Supplementary Figs. 28–35 and Fig. 4f, the surface charge density of posi-PTFE, posi-PFA, posi-PET, posi-PVC, and posi-PP decreased by 8.58–15.16% after 5000 repetitions of contact-separation, while only 45.55%, 53.60% of the initial value were maintained for posi-Si rubber, posi-PE is. Therefore, the positive surface charge storage capacity of PTFE, PFA, PVC, and PET is superior to that of PP, PE, Si rubber, Nylon and glass (Fig. 4g).

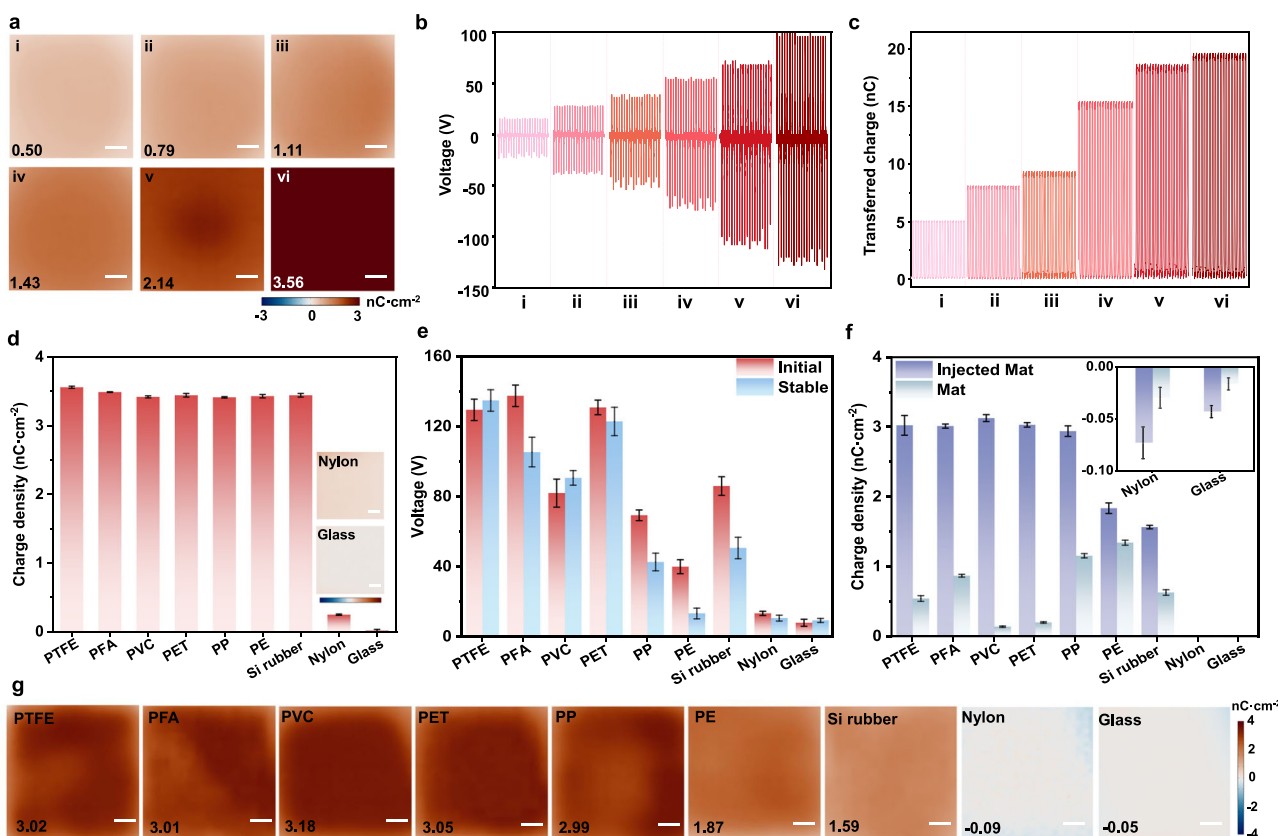

**Fig. 4 | Positive surface charge tuning and standardization for triboelectric materials. a** Tuning and standardization of positive surface charge on PTFE (Scale bar: 3 mm). **b, c.** The (**b**) output voltage, (**c**) transferred charge ($Q_{sc}$) of posi-PTFE based TENG injected with different positive surface charge density ($2 \times 2$ cm², $F = 50$ N). **d** The positive charge injection capability of various materials (distinct samples, mean ± s.d., $n = 5$. The inset images demonstrate the surface charge density of Nylon and glass, scale bar: 3 mm, color bar: −4 nC cm⁻² - 4 nC cm⁻²). **e** The triboelectric output voltage of various materials with injected positive charge (same sample measured repeatedly, mean ± s.d., $n = 5$. **f** The interfacial electrostatic positive charge transfer properties of various materials (same sample measured repeatedly, mean ± s.d., $n = 5$. The inset demonstrates the partially enlarged charge density of Nylon and glass). **g** The residual surface charge distribution of various materials (Scale bar: 3 mm).

Overall, the proposed imaging method combined with the uniform ion injection technique are effective in visualizing and regulating surface charge for a variety of triboelectric materials. Accordingly, the high-performance TENG was demonstrated by identical material with surface charge difference, and the polarity of tribo-negative materials could be switched to positive by charge tuning. This opens up more possibilities for the option of triboelectric materials and refining the triboelectric series.

## Surface charge dissipation properties and mechanism

In order to ensure the long-term durability of TENG, it is preferable for the accumulated surface charge to remain intact for an extended period. Typically, a triboelectric material with superior charge storage property to minimize charge dissipation is significant. To this end, the stability of both negative and positive charges injected into various polymer films was also measured. Figure 5a and Supplementary Fig. 36 show that the initial surface charge density of -3.02 nC cm$^{-2}$ in neg-FTFE decreased to −2.83 nC cm$^{-2}$ within 14 days, and subsequently remained steady. Only a little dissipation of the surface charge occurred at the film edge even after 140 days. The observed loss in

surface charge density amounted to 9.95%, confirming its superior charge storage capacity. For other negatively charged materials (Fig. 5b and Supplementary Figs. 37–43), 92.88% of the injected negative charges on PFA were maintained after 70 days, verifying that the materials with abundant fluorine atoms can trap the negative charge stably. As for PVC, PET, and PP, a 10.08%, 17.15%, and 19.15% loss of negative surface charge density were found after 70 days (Fig. 5b). In comparison, the negative charge storage capacities of PE, Si rubber, and Nylon are inferior, which appear to be uniformly dissipated (Supplementary Figs. 41–43). The surface charge density on PE quickly decayed to 76.67% and 41.91% of the initial value within 1 day and 7 days, respectively. The deposited negative charge on Si rubber and Nylon cannot maintain 1 h. Consequently, the negative surface charge storage capacity is ranked as follows: PTFE, PFA > PVC > PET > PP > PE > Si rubber > Nylon, which is firstly visualized to verify the quantified triboelectric series reported[23].

The positive charge dissipation properties of various polymer films were also investigated, as shown in Fig. 5c, d and Supplementary Figs. 44–51. Specifically, the surface charge density on PTFE decayed -5% after 140 days, similar to that of PFA (loss of 7.13%). This confirmed

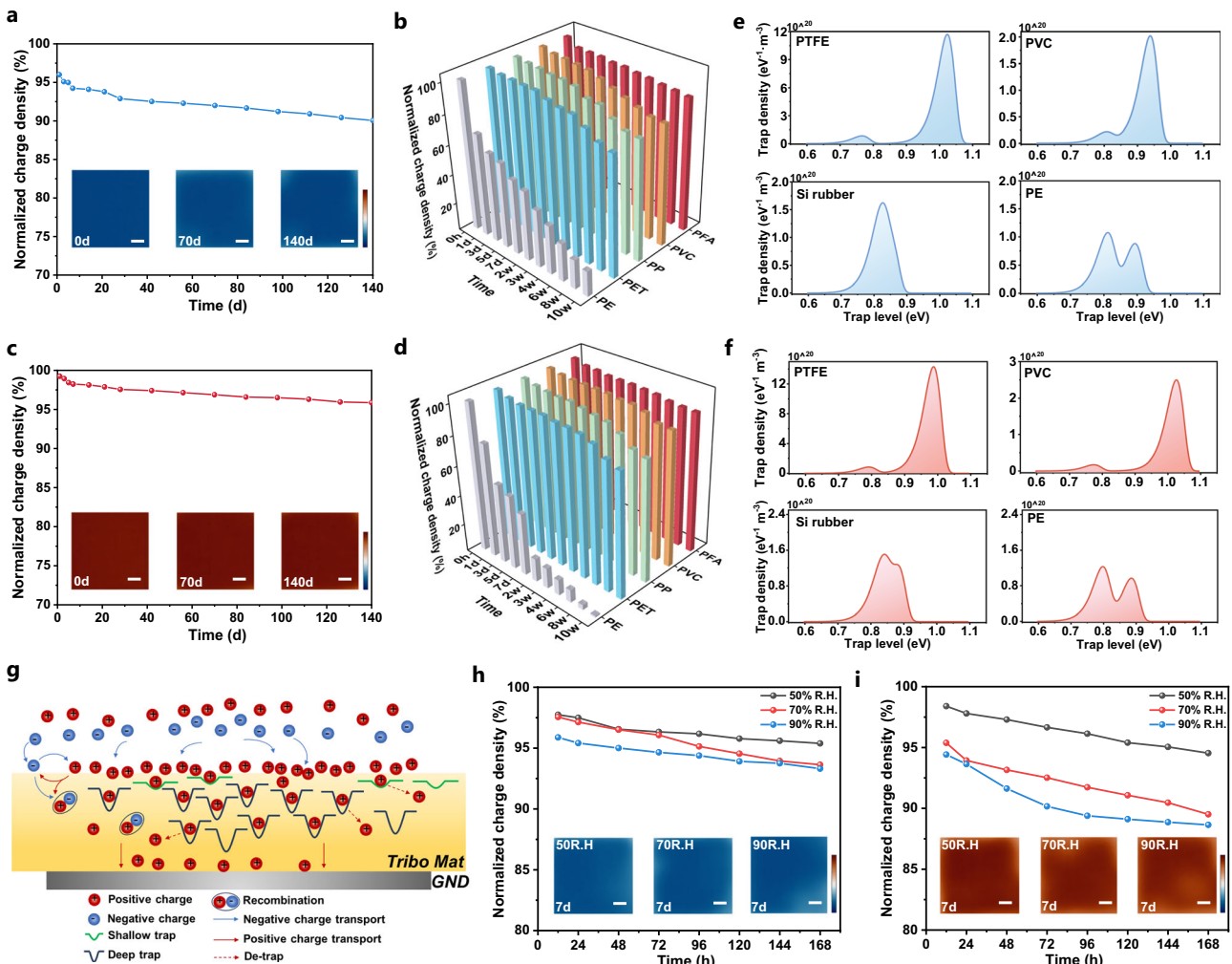

**Fig. 5 | Surface charge dissipation properties and mechanism. a** Decay of the negative surface charge on the PTFE (The inset images demonstrate the surface charge density of PTFE at 0d, 70d and 140d, scale bar: 3 mm, color bar: −4 nC cm$^{-2}$ ~ 4 nC cm$^{-2}$). **b** Comparison of the negative surface charge dissipation properties of various materials. **c** Decay of the positive surface charge on the PTFE (The inset images demonstrate the surface charge density of PTFE at 0d, 70d and 140d, scale bar: 3 mm, color bar: −4 nC cm$^{-2}$ ~ 4 nC cm$^{-2}$). **d** Comparison of the

positive surface charge dissipation properties of various materials. **e** The electron trap level distribution of PTFE, PVC, Si rubber and PE. **f** The hole trap level distribution of PTFE, PVC, Si rubber, and PE. **g** Mechanism of the surface charge dissipation on polymer surface. **h-i** The influence of relative humidity on (**h**) negative and (**i**) positive surface charge decay process of PTFE (The inset images demonstrate the surface charge density of PTFE at 50% R.H., 70%R.H. and 90%R.H. after 7d, scale bar: 3 mm, color bar: -4 nC cm$^{-2}$ ~ 4 nC cm$^{-2}$).

that the tribo-negative materials regulated to positive ones could maintain stability, which is superior to the stability reported previously based on FEP films[20]. The difference might be ascribed to the role of high electrical field that rivets positive charges into deep traps. In addition, the positive surface charge density of PVC, PET, and PP maintained 89.91%, 82.85%, and 80.75% of the initial value after 70 days, respectively. PE, Si rubber, and Nylon demonstrated poor positive charge storage capabilities and cannot preserve positive charge for 7 days or even one hour, as shown in Supplementary Figs. 49–51.

Generally, the material contains various structural and chemical imperfections inside the dielectric, which create carrier traps (e.g., electron and hole traps)[37]. Carrier traps belong to localized states with energy levels in the bandgap and are recognized as critical charge storage containers, which can be categorized into two distinct types: deep traps and shallow traps. The schematic diagram of their distribution in the energy band structure is shown in Supplementary Fig. 52. Electron traps (electron/negative ion capture sites) and hole traps (positive ion capture sites) are distributed on both sides of the Fermi level. Using hole traps as an example, the shallow traps are closer to the bottom of the valence band, indicating a relatively low energy need for the carrier residing in it. On the contrary, the deep traps are situated away from the valence band and located near the Fermi level, and the energy required for the carrier to escape is high[38]. The positive/negative charges injected into the dielectric material through corona discharge are captured by both the shallow and deep traps, and the stored charges in the traps will undergo dissipation when the applied voltage is removed. In general, charge carriers in shallow traps will dissipate first, while carriers in deep traps will be stored longer.

Figure 5e, f and Supplementary Figs. 53 and 54 depict the electron and hole trap distributions of the various dielectric materials obtained by the isothermal surface potential decay (ISPD) method. The two characteristic peaks in the spectrum represent the shallow (lower energy level) and deep traps (higher energy level). We can find that all carrier traps (electron and hole traps) in PTFE, PFA, PVC, PET, and PP belong to deep traps. Specifically, both the electron and hole deep trap density of PTFE, PFA reached the order of $10^{21} eV^{-1} m^{-3}$ at -1.0 eV, which is ~5 times higher than that of PVC and PET. As a result, it is difficult for the charge carriers to escape once they enter the PTFE and PFA traps, indicating the superior positive, negative charge storage and retention capabilities. However, the bulk of electron and hole trap energy levels in silicone rubber, Nylon and glass mainly occur at 0.8 eV and 0.7 eV, respectively, indicating the shallow traps are the dominant carriers and the electrons or ions are easy to escape. Thus, the carriers produced by corona discharge cannot be stored in these materials and dissipate quickly. In the case of PE, the density of shallow traps exceeds that of deep traps, resulting in the gradual dissipation of surface charges over time. In short, positive charges disperse faster than negative charges due to the lower densities of electron traps compared to hole traps. Hence, the variation in surface charge dissipation characteristics among different triboelectric materials can be ascribed to the inherent qualities of carrier trap distribution. Enhancing the energy level and density of deep traps in dielectric materials is essential for maximizing the surface charge storage capacities, which is highly valuable for the advancement of high-performance triboelectric materials and devices.

The trapping and de-trapping phenomena of carriers commonly determine the intrinsic charge dissipation characteristics of the material. According to current consensus, the surface charge mainly dissipates in the following three ways, (1) dissipation through the interior of the dielectric material (bulk conduction), (2) dissipation along the surface of the dielectric material (surface conduction), or (3) neutralization with charged particles in the gas or air. Typically, these three pathways exist simultaneously, as illustrated in Fig. 5g. When the surface charge is mainly dissipated through the bulk conduction of dielectric materials, the shape of the surface charge distribution remains unchanged and the dissipation rate remains constant throughout. For triboelectric materials with higher bulk conductivity (such as silicone rubber, PE), it was found that both the positive and negative charge densities decrease uniformly across the observation periods (Supplementary Figs. 41 and 42 and 49 and 50), indicating that the bulk conduction predominates the dissipation process. The surface charge distribution morphology will change when pathways 2) and 3) are the primary for dissipation[39]. The negative surface charge of PTFE, PFA, PVC, PET, PP remained evenly distributed throughout a month (4 weeks), and no surface charge diffusion was observed towards the surrounding area (Supplementary Figs. 36–40 and 44–48). After 6 weeks of dissipation, the charge density at the edge of the sample experiences a modest reduction, however the overall charge distribution remains predominantly uniform. Regarding positive charges, a slight decrease in the PFA, PVC, PET, and PP surface charge density around the edge area was found after 3 days, indicating that the contribution of surface conduction in the positive charge dissipation process is slightly larger than that of negative charges. In general, the surface charge stability of PTFE, PFA, PVC, PET, and PP is better, and surface conduction is not the primary dissipation channel as these insulation materials possess low surface conductivity.

Additionally, as TENG's output performance is humidity-sensitive, the impact of relative humidity (R.H.) on surface charge dissipation was further investigated. As shown in Fig. 5h and Supplementary Fig. 55, the negative surface charge density of PTFE remained comparatively stable at 50% R.H., while the decay rate increased at 70% R.H and 90% R.H. There is 4.61%, 6.37% and 6.69% loss of the surface charge density at 50%, 70% and 90% R.H. after 7 days, respectively. As for the positive polarity, the surface charge density has decreased by 5.47%, 10.49%, and 11.36% at 50%, 70%, and 90% R.H. after 7 days (Fig. 5i and Supplementary Fig. 56). Thus, the stability of positive surface charge is basically lower than the negative one at higher R.H. conditions. According to the surface charge distribution morphology given in Supplementary Figs. 57 and 58, surface charges are uniformly distributed within 4 weeks and there is no apparent edge diffusion at 30% R.H. After two hours of exposure to the 90% R.H. environment, both positive and negative surface charge density demonstrates a significant diffusion in the surrounding edge areas. The surface charge distribution becomes less uniform as the dissipation time increases, resulting in an uneven distribution after 4 weeks of exposure. Normally, the ionization process of gases is unaffected by the low electric field produced by the surface charges[40]. In other words, the humidity has no influence on the ion-pair generation from natural ionization in the gas under low-field conditions. However, water in the vapor phase will cause potentially conductive layers at interfaces. Under high humidity conditions, water molecules will create a micro "water film" to cover the sample surface, which enhances the surface conductivity and facilitates the neutralization of surface charges. The inhibition of surface lateral conduction by hydrophobic coatings, such as hexamethyldisilazane, has been verified as an effective strategy to avoid charge decay under harsh environments[41,42].

On the whole, both the negative and positive surface charge deposited by three-electrode induced corona discharge demonstrated outstanding stability, particularly for the dielectric materials with high-density deep traps like PTFE and PFA. The increase of deep trap density in dielectrics could enhance the surface charge storage capacity and reduce charge dissipation, which provides new inspiration and guidance for developing high-performance triboelectric materials and devices. The surface charge polarity tuning of traditional tribo-negative materials would create more tribo-positive materials. It is desirable to optimize charge injection parameters for various materials in order to construct a material database with adjustable triboelectric characteristics and adaptability to varied environments.

**Application of surface charge visualization and tuning for TENG**

The proposed VSQ method and single polarity charge deposition technique endow surface charge visualization and quantification for triboelectric materials, providing a strategy to improve the output performance as well as status monitoring of TENG during long-term operation. Typically, CE performance degradation exists in TENG owing to mechanical wear and structural failure (protrusion, depression, etc.) caused by continuous operation. Visual reconstruction analysis of the surface charge distribution in the triboelectric layer can be utilized to clarify the intrinsic correlation between macroscopic CE performance and surface charge density under various conditions, as well as device status evaluation and fault diagnosis. Here, various concave defects of varying sizes were constructed in the triboelectric layer, which can be identified by surface charge visualization. As shown in Fig. 6a, the charge density at the defect edge is greater than that in the middle region, and this difference becomes more pronounced as the size of the defect increases from 1.5 mm to 5 mm, indicating that concave defects would not be easily contacted during TENG operation. Additionally, we constructed the patterned defects on Si rubber as a triboelectric layer to evaluate its surface charge distribution after repeated contact with Aluminum. Figure 6b presents that letters "T E N G" with high resolution in surface charge difference can be identified, revealing the existence of the defects. Thus, the proposed surface charge visualization method is capable of detecting surface defects and assessing interface charge for TENG.

Triboelectric materials with high surface charge density are promising for TENG-based self-powered contact and non-contact sensing applications. We designed a single-electrode TENG consisting of a positively charged PTFE triboelectric layer and a sputtered gold electrode. The device was mounted on the finger of a manipulator to realize contact perception. As shown in Fig. 6c, the gentle tapping (force <1 N) of index finger on the object surface produced a relatively strong voltage response. For instance, the voltage response of the manipulator towards PVC, PFA, PET, PE, PP, and Nylon achieved −51 V, −46 V, −36 V, −30 V, −28 V and −17 V, respectively. The promising material identification is expected to be accomplished according to the response difference. The merit lies in the high surface charge density-based sensing layer with high-sensitive response to a tiny pressure.

Besides, the high surface charge density also facilitates the construction of high-performance non-contact TENGs. Here, PTFE has been verified to possess exceptional capacity for tuning positive and negative surface charges with enduring stability. We further employed the posi-PTFE/PTFE and neg-PTFE/PTFE as triboelectric layers for non-contact TENG with an initial thin gap of 0.2−0.5 mm to highlight its application potential as e-skin for non-contact sensing. According to Fig. 6d and Supplementary Figs. 59 and 60, the output voltage and transferred charge reached 45 V, and 15 nC with 0.2 mm gap, and then attenuated at 0.5 mm owing to the weakened electrostatic induction. However, the surface charge density of the charged layer remained stable after operation, with the residual charge density for different gap of 3.03 nC cm$^{-2}$, 3.30 nC cm$^{-2}$, and 3.38 nC cm$^{-2}$. In addition, higher approach-separation frequency delivered larger output voltage as the electrostatic induction happens more quickly (Fig. 6e), based on the consistent total transferred charge at 1−5 Hz (Supplementary Figs. 61 and 62). This non-contact device also delivers long-term output stability, even after 3000 s of continuous operation (Supplementary Fig. 63).

Further, the application scenarios for positive polarity PTFE were demonstrated by fabricating the posi-PTFE based non-contact TENG as e-skin for self-powered sensing. The devices were mounted on five fingers of a manipulator to construct a sensor array. A single sensing unit (size: 1.5 × 0.8 cm) can produce an output voltage of ~3.5 V when the distance is less than 2 mm (Fig. 6f and Supplementary Movie 2). Additionally, the manipulator with programmed motion was used to perceive polyhedral structures without physical contact. As illustrated

in Fig. 6g, typically, the approach-separate away (repeat non-contact grabbing) to the square will produce five similar voltage signals since all of the fingers were in close proximity to the item (Supplementary Movie 3). As for the sphere, the voltage generated by the middle finger and ring finger is larger than the others. Moreover, being closest to the cylinder's bottom surface, the thumb finger produced the maximum voltage signal compared to the other four fingers. Little response signals to the cone are found as all the fingers are relatively far from the object. Besides, the perception of placement direction of a polyhedron was realized, as shown in Fig. 6h and Supplementary Movie 4. Specifically, the forefinger, middle finger, ring finger, and little finger will generate similar responses when the triangular base is close to them, and the edge close to the thumb has almost no signal output. The signal is opposite when only the triangle's base is close to the thumb. The middle finger's with highest response voltage indicates that the triangle's base is facing outward. This high sensitivity of the non-contact manipulation is attributed to the high surface charge of PTFE enabled by regulatable charge injection. The strategy is promising for the development of diversified e-skins for human-machine interfaces (HMI), smart robotics, etc.

## Discussion

In summary, we achieved the visualization and standardized quantification of surface charge based on the proposed VSQ method. The two dimensional (2D)/3D surface charge distribution was inverted using the surface potential measured by the electrostatic probe, and the method demonstrated high precision and robust noise immunity. Meanwhile, a three-electrode-induced corona discharge for single polarity charge injection was demonstrated, which is promising for triboelectric charge polarity tuning and triboelectric properties modification of various materials. Mechanism of the stabilization and dissipation of surface charges reveals that materials with high electron/hole deep trap density deliver superior charge storage capacity and long-term stability. Based on an identical material, a single electrode mode and a non-contact mode TENG composed of the posi-PTFE/PTFE triboelectric pair was demonstrated to produce high output voltage of 136 V, and 50 V, respectively. The surface charge VSQ method is promising for detecting defects and monitoring material status for TENG, which presents important potential for refining the triboelectric series, as well as advancing the development of high-performance triboelectric materials and TENGs.

## Methods

### Sample preparation and charge injection

PTFE, PFA film was supplied by Cleverflon Technology (Jiangsu, China). PVC, PET, PP, PE, Nylon, Silicone rubber and glass was provided by Guanmei Plastic (Guangdong, China). The PTFE, PFA, PVC, PET, PP, PE, Nylon, Silicone rubber, glass with a thickness of 0.1−0.3 mm were cut into 20 × 20 mm film, which was adhered to a conductive textile (Ni-fabric) tape (20 × 20 mm) as the electrode. The other side of the conductive textile was arranged on the acrylic board (30 × 30 mm) substrate by its own adhesive. The initial surface charge on the sample was removed by an air-ionization gun for 60 s before charge injection.

The three-electrode system composed of the tip (curvature radius: 0.20 mm), metallic grid (electrode radius: 25 mm, screen radius: 0.5 mm), and plane electrode (Side length: 10 cm) is made of brass or stainless steel. The tip-grid electrode and grid-plane electrode were spaced apart by 1 cm and 0.2 cm, respectively. For positive charge deposition, a high voltage of 8.5 kV, 2 kV was supplied to the tip and grid electrode, whereas the value was set to −8 kV, 2 kV for negative charge deposition. The sample was placed between the grid and the grounded plane electrode. The single polarity charge deposition duration time is set to 5 min. Besides, lower surface potential/charge control is achieved by adjusting the corona discharge parameters and duration time, as discussed in Supplementary Note 8. The dust figure

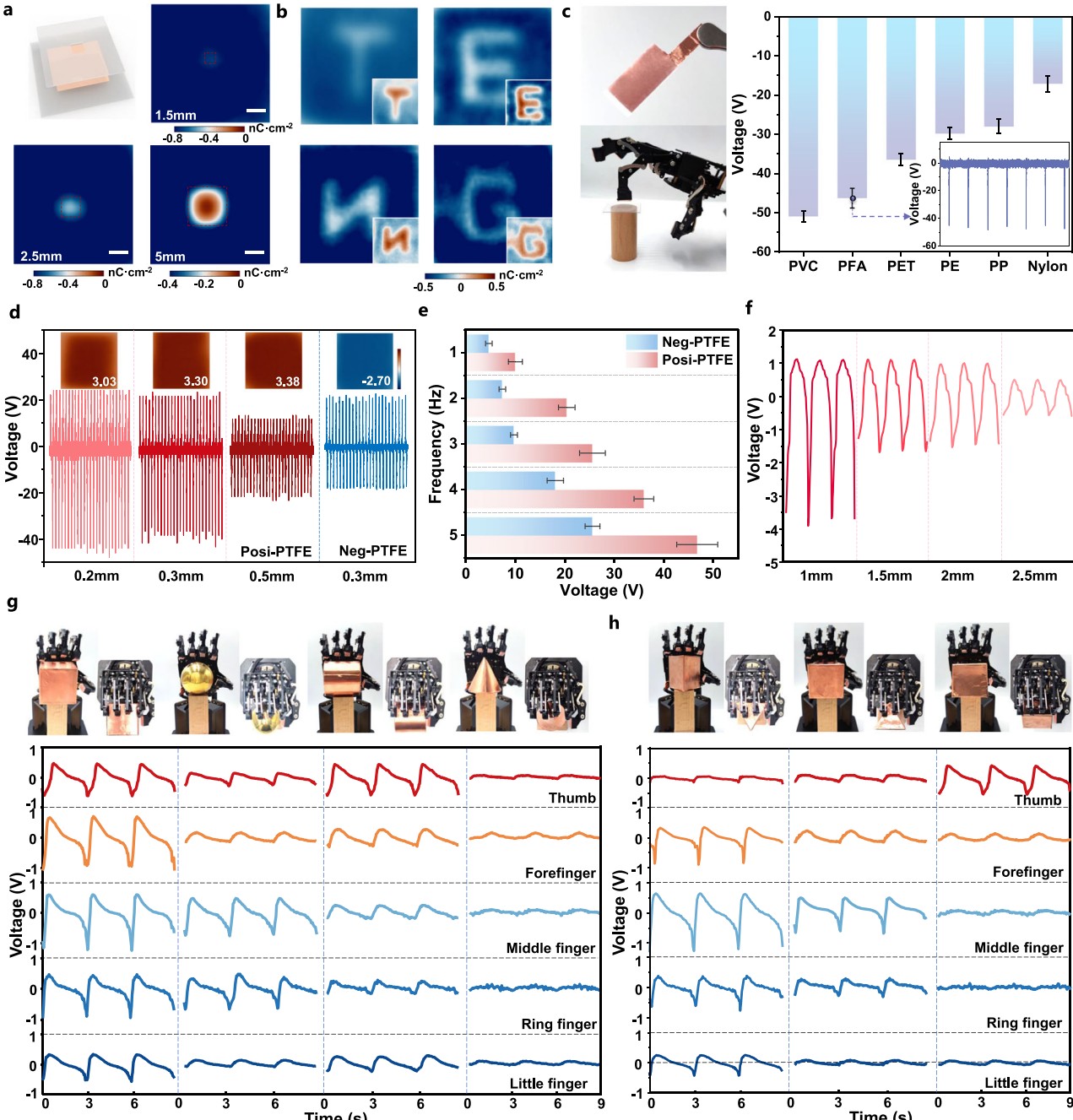

**Fig. 6 | Application of surface charge visualization and standardization for TENG. a** Surface depression identification of triboelectric material enabled by the surface charge standardized imaging method (Scale bar: 3 mm). **b** High-resolution of the standardized imaging method enables the identification of "TENG" letter composed of interface defects. The inset image demonstrates the charge density distribution with high contrast. **c** Posi-PTFE based single electrode TENG mounted on manipulator for contact perception and the relevant voltage response to various materials. (same sample measured repeatedly, mean ± s.d., $n = 5$.) The inset image demonstrates the response signal of the e-skin. **d** The output test was conducted by powdering printer toner particles (Static Control, LT1821K, blue) through an air blower, and detailed information can be found in Supplementary Note 5.

voltage of posi-PTFE/PTFE, neg-PTFE/PTFE based non-contact TENG at different gap distances. The inset images demonstrate the surface charge density of the non-contact TENG after the tests, color bar: −4 nC cm⁻² - 4 nC cm⁻² **e** The influence of approach-separation frequency on the output performance of non-contact TENG. (same sample measured repeatedly, mean ± s.d., $n = 5$.) **f** The voltage response of manipulator finger mounted with posi-PTFE based TENG at different distances towards the items. **g** Contactless response of manipulator's finger towards different polyhedrons. **h** Contactless response of manipulator's finger towards triangular prisms in different arrangements.

test was conducted by powdering printer toner particles (Static Control, LT1821K, blue) through an air blower, and detailed information can be found in Supplementary Note 5.

## Fabrication of TENG

Contact separation TENG: Triboelectric film ($20 \times 20$ mm), such as PTFE, PFA, etc. was adhered to the acrylic substrate by a conductive

textile tape based electrode. Then, the positive or negative surface charge was deposited on the triboelectric layer. The other triboelectric film was attached to the moving end of linear motor by the grounded conductive textile tape. The gap between the triboelectric pair was fixed at ~3 mm. For the non-contact TENG, the PET ($2 \times 20$ mm) with thickness of 0.2 mm, 0.3 mm, and 0.5 mm was attached around the acrylic substrate to construct the gap.

Single electrode TENG for manipulator: The copper electrode (5 mm × 15 mm) was pasted on one side of the PTFE. The other side of PTFE with positive charge deposition was utilized as a triboelectric layer. Then, the single electrode TENG was adhered to the finger of the manipulator by the Very high bond (VHB) tape.

## Measurement

Surface potential distribution was measured by the electrostatic meter (Trek 341B, measurement range of -19.99 kV to +19.99 kV, maximum measurement error of ± 0.1%) based 2D scanning detection system. The probe was grasped by an insulated clamp, whose movement was driven by two stepping motors. The scanning step for surface potential was set to 2 points per mm for the x-axis and y-axis and the scanning grid was 30 mm × 30 mm (a total of 3600 points). The gap distance between the sample surface and the probe was 2 mm. The obtained data was stored by the oscilloscope (Tektronix DPO 4102B).

The contact-separation process of TENG was driven by a linear motor (R-LP3) with controllable force and frequency. The electrometer (Keithley 6517B) connected with the multimeter (Keithley 6510) was employed to measure the transferred charge, and the output voltage was recorded using an oscilloscope. All measurements were made under ambient conditions (25 °C, 30–40% relative humidity). The charge dissipation experiments were carried out in a sealed room with a regulated relative humidity at 25 °C. Trap density measurement and calculation were carried out based on ISPD, with detailed information given in Supplementary Note. 9 and Supplementary Fig. 64. Moreover, a programmable manipulator (Hiwonder, uHandPi 2.0) was employed to conduct relevant demonstrations. Specifically, the five fingers and gimbal of the bionic manipulator are driven by the Hiwonder LFD-01 and LD-1501MG digital servo motor. The controller is based on an STM32 microcontroller. The PC control software (uHand V1.1) was developed using the Windows Presentation Foundation (WPF). The experiment involved manipulating the displacement and duration of the digital servo motor connected to five fingers to accomplish various movements. For example, the finger action duration is set to 3 s (gripping time: 1 s, recovery time: 2 s) for non-contact perception, and the relative displacement ratio (Corresponding bending angle) is 178 (thumb), 208 (index finger), 282 (middle finger), 255 (ring finger), 144 (little finger) compared to the initial displacement of 1500.

## Reporting summary

Further information on research design is available in the Nature Portfolio Reporting Summary linked to this article.

# Data availability

The data that support the findings of this study are available within this Article and its Supplementary Information. Raw data necessary to reproduce the figures within this Article are available in figshare under https://doi.org/10.6084/m9.figshare.25145585.

# Code availability

Extraction of the scanned surface potential was conducted by MATLAB R2022a. The core parameters (H-matrix, etc.) of charge visualization and standardized quantification methods was obtained by MATLAB R2022a and COMSOL Multiphysics 5.6. Analysis of surface potential and charge was conducted based on COMSOL Multiphysics 5.6. Code for data cleaning and analysis and the raw data of Case 1 and Case 2 are available in Github under https://github.com/roy1025/VSQ.

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

## Acknowledgements

We would like to thank Mr. Haocheng Deng, Mr. Shengyao Shi, Mr. Dazhi Su, Miss. Shiyi Mao, Dr. Zijun Pan, and Mr. Pu Han for their helpful discussions and measurements. The current research was supported by the National Science Fund for Distinguished Young Scholars (Grant No. 51925703) received by T. S., the National Natural Science Foundation of China (Grant No. 52207169) received by Y. L., the National Natural Science Foundation of China (Grant No.52103254, 52273244) received by J. X., and the Postdoctoral Fellowship Program of China Postdoctoral Science Foundation (Grant No. GZC20232635) received by Y. L.

## Author contributions

Yi Li, Yi Luo, Song Xiao, and Zhong Lin Wang conceived the idea and designed the experiments. Yi Li, Yi Luo, Song Xiao, and Cheng Zhang carried out the experiments, calculations, and analyzed the data. Yi Li, Yi Luo, and Cheng Pan contributed to the surface charge VSQ method. Zhaolun Cui, Ju Tang, Fuping Zeng, and Bangdou Huang commented on the results and mechanism analysis. Yi Li, Yi Luo, Song Xiao, and Jiaqing Xiong wrote the manuscript. Zhong Lin Wang, Jiaqing Xiong, Xiaoxing Zhang, and Tao Shao supervised this project. All the authors discussed the results and commented on the manuscript.

## Competing interests

The authors declare no competing interests.
