## [Peer Review File · Nature Communications]

Visualization and Standardized Quantification of Surface Charge Density for Triboelectric MaterialsReviewer #1 (Remarks to the Author):

In the manuscript titled "Visualization and Standardization of Surface Charge Density for Triboelectric Materials", the authors presented several key innovations in the field of triboelectric nanogenerators (TENGs), which are devices that generate electricity from mechanical motion through contact electrification and electrostatic induction. The authors introduce a method for directly visualizing surface charge distribution on triboelectric materials using the electrostatic surface potential measured by the Kelvin probe. They also propose an efficient tuning strategy for single polarity surface charge based on corona discharge with a three-electrode design. This work uncovers the long-term stability, storage, and dissipation mechanisms of injected charges, both negative and positive.

One notable finding of this work is an approximately 70-fold enhancement in output voltage for TENGs fabricated with identical triboelectric materials after charge injection. This results in a stable surface charge with only a 5% decay after 140 days. The authors also demonstrate the application of charged PTFE-based TENG as robotic electronic skins for non-contact perception of object geometries.

One of the most interesting and important contributions of this work is the development of a spatially resolved Kelvin probe mapping tool (and retrieval procedure) for large-size samples with high surface potential values. This tool is much-needed in a sense that the current Kelvin probe force microscopy methods (based on atomic force microscope) cannot measure a large sample scale of millimeters, nor can it handle a large value of surface potential > 10 V (although the high spatial resolution of AFM based KPFM does not need the retrieval procedure because the spatial resolution is high enough). In this work, the authors developed a tool and an algorithm to retrieve the surface potential map of large scale high voltage sample.

Besides the innovation in measurement, the manuscript details the methods for surface charge tuning of various polymers, offering a more refined standardization strategy for the triboelectric series of materials.

Overall, this research provides essential tools for surface charge visualization and standardization, presenting a new approach for high-performance TENGs, which have broad applications in energy harvesting and self-powered sensing.

These contributions indicate a significant advance in triboelectric material research with the potential of further enhancing this type of material for energy harvesting. I recommend the publication of this manuscript.

Reviewer #2 (Remarks to the Author):

This manuscript presents a method for visualizing and standardizing the surface charges of triboelectric materials through a technique called surface charge standardized imaging. This method not only enables the visualization and quantification of surface charge distribution but also allows for the use of a three-electrode system for the injection of single-polarity charges. Furthermore, the authors propose theoretical equations related to the linear superposition of the effects of all surface charges. However, detailed information about the results and experiments is required for the better understanding of readers. Additionally, the objectives for utilizing a robotic electronic skin (e-skin) are not sufficiently clear. Therefore, I recommend the publication of this work in Nature Communications after revising the manuscript with following comments.

1. Please define 'e-skin' and discuss the motivation for using the surface charge standardized imaging method to detect surface defects in the 'Introduction' section.
2. In Fig. 1g, how does a PTFE-PTFE couple (i.e., identical material) produce output voltage? What determines whether PTFE is tribo-positive or tribo-negative? Generally, triboelectricity is generated between relatively tribo-positive and tribo-negative materials.
3. Please add scale bars in Supplementary Fig. 9 and other visual for a clearer understanding of uniformity.
4. Please add a caption what Figs. 2e(i)-(vi) stand for. How is the negative surface charge adjusted? Is it a result of varying deposition duration time?

5. In Fig. 3g, why do Nylon and glass exhibit lower positive charge acceptances than other materials?
6. In Figs. 4f and 4g, what causes the difference between a small peak of PTFE at ~ 0.8 eV and the large peak at ~ 1.0 eV?
7. On page 21, the paragraph about a robotic e-skin is somewhat unclear. Please clarify why both positive- and negative-PTFE are used for e-skin and the research objectives of robotic experiments.
8. Please describe the operational method of the robotic e-skin (module circuit, software, experimental condition, etc.) in the 'Methods' section.

Reviewer #3 (Remarks to the Author):

This manuscript chapter presents an approach to visualize and calculate surface charges in triboelectric nanogenerators using surface potential measurement. It links such measurements and charge extraction methods to enhancing performance in energy harvesting and self-powered sensing applications from triboelectric materials. The study introduces a surface charge imaging method using electrostatic surface potential measured by a Kelvin probe, coupled with an efficient tuning strategy for a single polarity surface charge based on corona discharge with a three-electrode design. This method allows visualisation and quantification of surface charge distribution on triboelectric materials.

There are some innovative aspects to the research presented, like the mathematical algorithm to go from surface potential to surface charge.

The research also shows a 70-fold enhancement in output voltage for triboelectric polytetrafluoroethylene (PTFE), offering stable surface charge with minimal decay over time. Additionally, the charged PTFE is applied as a robot electronic skin for non-contact perception of object geometries, which is certainly an interesting application.

Some of these results will certainly be of interest and novelty to the scientific community in tribology and electrostatic nano-harvesters. However, I have serious concerns for the quality and representations given in this manuscript, and therefore I don't recommend for publication/ For example, while it is of use to have an accurate calculation of charge from surface potential, there is no detailed exploration of the accuracy of the method proposed against simulations. Kelvin probe data can be very easily simulated in packages like COMSOL or ANSYS, allowing checking computational precision of the proposed maths for converting potential to charge. This seems to the trust of the first part of the paper, but is purely a maths-filled jargon without any corroboration that the steps taken to improve accuracy indeed work, beyond the minimisation of error in the equations.

The stability of positive charges compared to negative ones in various environmental conditions is mentioned but not deeply analysed. In fact, a large part of the literature in the field that already addresses this challenges has been entirely ignored. Several works link charge degradation to water presence on the materials, and lateral conduction, indicating that hydrophobic coatings can work as a solution to avoid charge decay. Here the authors merely changed the materials, but there is not real evidence of what is causing the degradation, and how a change in material improves it. See for example works in <https://doi.org/10.1109/IBERSENSOR.2014.6995520>, <https://doi.org/10.1109/94.660767>, <https://doi.org/10.1109/PVSC.2014.6924985>

Some aspects do not seem to be immediately relevant and sufficiently supported:

- Line 28, why is the visualisation and standardisation of surface charges essential for high-performance nanogenerators? – it is clearly important, but certainly not a prerequisite. In fact, the text itself later on described the charge density as the primary requirement for improved performance.
- Line 82: "However, the ion deposition process is random and cannot be visualised or standardised. The needle-plate electrode": this is incorrect, as can be seen from work in references doi.org/10.1016/j.apsusc.2017.03.204, doi.org/10.1016/j.apsusc.2006.03.075, doi.org/10.1016/j.polymdegradstab.2014.03.017, doi.org/10.1016/j.egypro.2016.07.090
- What the authors refer to as standardisation is entirely unclear. It appears they are misusing the word, or have a different meaning intended.
- A large part of the argument is based around triboelectric materials, but all experiments are conducted using corona discharge, which uses a different mechanistic approach to building charge

concentration on a surface.

- Line 183: "Accordingly, we proposed the adaptive weight generalized cross validation (AWGCV) to select the regularization parameter η , as illustrated in Fig. 1c iii." – This figure is merely showing a set of equations, which are already described in the text, to exemplify the iteration flow of finding a right approach to solving the linear system, but it does not say anything about why, how, and the advantages to using AWGCV.

Also, another crucial questions have not been addressed:

If the distance between the probe and the sample changes (as it would be the case in Kelvin probe)) the transfer matrix H in their algorithm would change. So does this render the algorithm only useful if using exactly the same machine with the same parameters?

It is unclear what Fig 1.a (first image) is supposed to be communicating. Not covered in the text, but it also just seems like a gimmick rather than a scientific schematic of relevance.

The units and magnitudes in the colour bars are not readable.

The images that portray real data do not have a lateral scale bar.

Finally, the level of scientific English in this article is also unsatisfactory:

- Line 26 is grammatically incorrect. Missing an article.

- Line 52

- Line 53, by a direct visually.

- And many others.

Dear Reviewers,

We appreciate your efforts in thoroughly reviewing our manuscript entitled “Visualization and Standardization of Surface Charge Density for Triboelectric Materials” (NCOMMS-23-51061). According to your valuable comments and suggestions, we have made extensive modifications to our manuscript and resubmitted a revised version for your consideration. The revisions for addressing the reviewers’ comments have been **highlighted in yellow** in the revised manuscript. Other modifications about discussion and language have been made to improve the manuscript, which have been **highlighted in red** in the revised manuscript. The point-by-point responses to reviewers have been listed as following. Thank you very much for your time and consideration.

Sincerely,

Zhong Lin Wang

Professor, Ph.D

Beijing Institute of Nanoenergy and Nanosystems| Chinese Academy of Sciences

Beijing 100083, People’s Republic of China

School of Materials Science and Engineering| Georgia Institute of Technology

Atlanta, Georgia, 30332-0245, United States of America

E-mail: wangzhonglin@binn.cas.cn

Jiaqing Xiong

Professor, Ph.D

Innovation Center for Textile Science and Technology | Donghua University

2999 North Renmin Road, Shanghai 201620, People’s Republic of China

Tel: (86)-19117305006 | E-mail: jqxiong@dhu.edu.cn

Xiaoxing Zhang

Professor, Ph.D

Key Laboratory for High-Efficiency Utilization of Solar Energy and Operation Control of Energy Storage System| School of Electrical and Electronic Engineering| Hubei University of Technology

Nanli Road, Wuhan 430068, People's Republic of China

State Key Laboratory of Power Grid Environmental Protection| School of Electrical Engineering and Automation| Wuhan University

Bayi Road, Wuhan, Hubei 430072, People's Republic of China

Tel: (86)-13627275072 | E-mail: zhangxx@hbut.edu.cn

Tao Shao

Professor, Ph.D

Beijing International S&T Cooperation Base for Plasma Science and Energy Conversion| In-stitute of Electrical Engineering| Chinese Academy of Sciences

No. 6, North Ertiao, Zhongguancun, Beijing 100190, People's Republic of China

Tel: (86)-13810312461 | E-mail: st@mail.iee.ac.cn

Point-to-point responses

The revisions in response to the Reviewers' comments are highlighted in yellow in the revised manuscript.

We have also made careful revisions to improve the manuscript. All the relevant changes are marked in red in the revised manuscript.

Supplementary Note 1, 4-8. Supplementary Fig. 1-6, 7, 9, 11-14, 16, 18-19, 22-53 for refining the manuscript were supplemented or updated in the Supporting Information.

Author's Response to Reviewer 1

Reviewer #1: In the manuscript titled "Visualization and Standardization of Surface Charge Density for Triboelectric Materials", the authors presented several key innovations in the field of triboelectric nanogenerators (TENGs), which are devices that generate electricity from mechanical motion through contact electrification and electrostatic induction. The authors introduce a method for directly visualizing surface charge distribution on triboelectric materials using the electrostatic surface potential measured by the Kelvin probe. They also propose an efficient tuning strategy for single polarity surface charge based on corona discharge with a three-electrode design. This work uncovers the long-term stability, storage, and dissipation mechanisms of injected charges, both negative and positive.

One notable finding of this work is an approximately 70-fold enhancement in output voltage for TENGs fabricated with identical triboelectric materials after charge injection. This results in a stable surface charge with only a 5% decay after 140 days. The authors also demonstrate the application of charged PTFE-based TENG as robotic electronic skins for non-contact perception of object geometries.

One of the most interesting and important contributions of this work is the development of a spatially resolved Kelvin probe mapping tool (and retrieval procedure) for large-size samples with high surface potential values. This tool is much-needed in a sense that the current Kelvin probe force microscopy methods (based on atomic force microscope) cannot measure a large sample scale of millimeters, nor can it handle a large value of surface potential > 10 V (although the high spatial resolution of AFM based KPFM does not need the retrieval procedure because the spatial resolution is high enough). In this work, the authors developed a tool and an algorithm to retrieve the surface potential map of large scale high voltage sample.

Besides the innovation in measurement, the manuscript details the methods for surface charge tuning of various polymers, offering a more refined standardization strategy for the triboelectric series of materials. Overall, this research provides essential tools for surface charge visualization and standardization, presenting a new approach for high-performance TENGs, which have broad applications in energy harvesting and self-powered sensing.

These contributions indicate a significant advance in triboelectric material research with the potential of further enhancing this type of material for energy harvesting. I recommend the publication of this manuscript.

Response: We highly appreciate your positive feedback on this work. As mentioned, the core innovation or progress of this work is concentrated on the following points,

1) The surface charge visualization or imaging strategy based on the Kelvin probe scanned surface potential and charge inversion method. We have decoupled the relationship between surface potential and charge, and realized the standardization and quantification of high-voltage interface macroscopic charges in triboelectric materials. The proposed strategy is complementary to the AFM-based KPFM method, which could be employed in other fields for surface charge measurement and visualization.

2) The surface charge tuning for triboelectric materials. We designed a three-electrode system for uniform, quantified injection of single polarity charge (positive or negative) on triboelectric materials combined with the proposed visualization tool. Further, the role of carrier traps (shallow, deep traps) on surface charge storage and dissipation was also revealed. Relevant findings provide a new framework for refined standardization of triboelectric materials.

3) High surface charge density enabled diverse applications. We reported the identical triboelectric material based TENG with relatively high output performance for the first time. Meanwhile, the high surface charge density also enabled the non-contact TENG for spatial information perception.

Besides, we have further improved this work based on the constructive comments from the other reviewers. Thanks again for your approval and consideration.

Author's Response to Reviewer 2

This manuscript presents a method for visualizing and standardizing the surface charges of triboelectric materials through a technique called surface charge standardized imaging. This method not only enables the visualization and quantification of surface charge distribution but also allows for the use of a three-electrode system for the injection of single-polarity charges. Furthermore, the authors propose theoretical equations related to the linear superposition of the effects of all surface charges. However, detailed information about the results and experiments is required for the better understanding of readers. Additionally, the objectives for utilizing a robotic electronic skin (e-skin) are not sufficiently clear. Therefore, I recommend the publication of this work in Nature Communications after revising the manuscript with following comments.

Response: We highly appreciate your positive feedback and constructive suggestions. We have provided more detailed information on the visualization and standardized quantification strategy, especially the charge inversion methods, charge injection and storage stability mechanism in the revised manuscript. Besides, the objective for the utilization of e-skin was also explained. With remarkable modifications, we appreciate your kind reconsideration of this work.

1. Please define 'e-skin' and discuss the motivation for using the surface charge standardized imaging method to detect surface defects in the 'Introduction' section.

Response: Thanks for your suggestions. Electronic skin (e-skin) is expected as a bionic electronic interface to distinguish different shapes, structures and perceive changes to external stimuli, which has promising applications in smart robots, artificial prosthetics, etc ^[1]. TENG is a promising strategy to fabricate self-powered e-skin for various applications, including tactile sensing ^[2], biomechanical monitoring ^[3], human-machine interface ^[4]. Recently, the non-contact-TENG that operates based on the principle of electrostatic induction has been proposed for diverse applications ^[5,6]. As a proof of concept, we designed a TENG-based e-skin with high surface charge density triboelectric layer for non-contact shape perception. This demonstration is intended to deliver the potential application of triboelectric materials with highly stable surface charge density. We further clarified the objective for utilizing e-skin in the Introduction section.

Besides, the motivation for using the surface charge standardized imaging method to detect surface defects was also explained. Normally, the output of TENG highly relies on the surface charge density of triboelectric material. The defects such as mechanical wear, protrusions and depressions inevitably exist on the triboelectric layer, which causes contact electrification (CE) efficiency and output performance degradation. To this end, the proposed surface charge visualization and standardized quantification method could be utilized for TENG performance evaluation and status diagnosis. Moreover, the association between macroscopic CE performance and surface charge density could be clarified by this strategy. We have updated the relevant content in the revised manuscript. (Manuscript: Page 3, line 2-5, 21-25)

2. In Fig. 1g, how does a PTFE-PTFE couple (i.e., identical material) produce output voltage? What determines whether PTFE is tribo-positive or tribo-negative? Generally, triboelectricity is generated between relatively tribo-positive and tribo-negative materials.

Response: Thanks for your comments. The PTFE-PTFE couple was employed as the comparison group to demonstrate the role of surface charge density in triboelectric output. We found a relatively low output voltage of 1.84 V existed, although the residual surface charge on PTFE was eliminated before testing. The surface potential and charge density of the PTFE-PTFE couple after tests were also measured, as shown in Figure R1. It can be found that PTFE_1 generated the negative CE potential/charge (blue region), whereas the positive potential/charge existed in the corresponding position on PTFE_2 (the yellow region). Thus, PTFE_1 and PTFE_2 acted as the “tribo-negative” and “tribo-positive” materials in this case.

Figure R1. The **a**) surface potential and **b**) surface charge of PTFE-PTFE couple after CE output performance test. (Scale bar: 3 mm)

Actually, CE between identical materials has also been reported previously by some scholars. For example, *Apodaca et al.* found that this behavior appears in poly(propylene), poly(styrene), Teflon, poly(vinyl chloride), and poly(dimethylsiloxane) (PDMS), which seems to be generic to non-elemental insulators [7]. *Xu et al.* pointed out that this phenomenon originated from the “curvature effect”. That is, CE of two pieces of chemically identical materials results in concave surfaces being positively charged, and convex surfaces being negatively charged [8]. The surface energies of different curved surfaces would differ, resulting in a change in surface states. Electrons could transfer from one material surface to another identical one with shifted surface states upon contact, as shown in Figure R2. Thus, the existence of micro/minor curved surface breaks the symmetry between the two sides, changes the surface state energy levels, and causes an electron transition [9]. The relevant analysis is consistent with the experimental results in this manuscript. We have added relevant analysis in the revised manuscript and Supporting Information. (Manuscript: Page 13, line 23-26, Supporting Information: Supplementary Note 7)

Figure R2. Mechanism of CE between identical materials of different surface curvatures. **a-c)** charge transfer before contact, in contact, and after contact. **d-f)** corresponding explanation of the surface charge transfer by the surface state model [8, 9].

3. Please add scale bars in Supplementary Fig. 9 and other visual for a clearer understanding of uniformity.

Response: Thanks for your suggestions. We have added the scale bars in Supplementary Fig. 9 and other visuals in the revised manuscript and Supporting Information. The revised version of Supplementary Fig. 9 is shown in Figure R3. (Supporting Information: Supplementary Fig. 11)

Figure R3. The surface potential and charge distribution of tip-plane electrode-induced corona discharge at **a)** -6 kV and 6 kV, **b)** -8 kV and 8 kV and **c)** three-electrode system (Scale bar: 5 mm). The results verified that the grid electrode ensured uniform distribution of surface charge.

4. Please add a caption what Figs. 2e(i)-(vi) stand for. How is the negative surface charge adjusted? Is it a result of varying deposition duration time?

Response: Thanks for your comments. The control of surface potential/charge is achieved by adjusting the corona discharge duration time. In order to achieve a manageable corona discharge injection for an extended duration, the separation between the metallic grid electrode and the sample was adjusted to 5 mm, and the high voltage applied to the tip electrode was reduced by roughly 500 V (8 kV for positive injection, -7.5 kV for negative injection). Figure R4 gives the relationship between the corona discharge (negative or positive polarity) duration time and the surface potential of PTFE. The negative and positive surface potential progressively rises from -271 V and 286 V at 3 min to -1706 V and 1946 V at 15 min, respectively, indicating the surface potential/charge could be regulated by adjusting the injection duration time. The surface potential distribution in Figure R5 further confirms the homogeneously surface charge injection.

Figure R4. The relationship between the corona discharge duration time and the surface potential. **a)** negative polarity (-7.5 kV tip electrode, -2 kV grid electrode). **b)** positive polarity (8 kV tip electrode, 2 kV grid electrode).

It should be emphasized that since the intensity of corona discharge is affected by several factors, such as external voltage and electrode structure, there is a particular uncertainty in the number of particles produced during a specific period. Therefore, the actual surface potential/charge needs to be determined by measurement and inversion calculations. The three-electrode charge deposition method proposed in this manuscript is a controllable and uniform surface charge control strategy. It can achieve surface charge density tuning on triboelectric materials combined with the charge inversion algorithm. We have added captions in Figs. 2e(i)-(vi) and relevant content to the revised manuscript. (Manuscript: Page 16, line 7-11, Supporting Information: Supplementary Note 8)

Figure R5. The surface potential distribution of a) negative polarity corona discharge and b) positive polarity corona discharge at different deposition durations (Scale bar: 3mm).

5. In Fig. 3g, why do Nylon and glass exhibit lower positive charge acceptances than other materials?

Response: Thanks for your comments. Here, we evaluated the positive charge acceptance properties of nine typical triboelectric materials. The results indicate that glass and Nylon have inferior positive charge acceptance capabilities compared to the other materials, mostly due to variations in carrier trap density.

Figure R6. Schematic diagram of the distribution of shallow and deep traps for electrons and holes in tribo-dielectric material.

Specifically, carrier traps in dielectric material belong to localized states with energy levels in the bandgap. Carrier traps are divided into two types: deep traps and shallow traps. The schematic diagram of their distribution in the energy band structure is shown in Figure R6. Electron traps (electron/negative ion capture sites) and hole traps (positive ion capture sites) are distributed on both sides of the Fermi level. Using hole traps as an illustration, the shallow traps are positioned near the lower end of the valence band, signifying that the energy needed for the carrier in the shallow trap is minimal. However, the deep traps are situated away from the valence band and near the Fermi level, and the energy required for the carrier to escape from the deep trap is high. The positive/negative charges injected into the dielectric material through corona discharge are captured by both the shallow and deep traps, and the stored charges in the traps will undergo dissipation when the applied voltage is removed. In general, charge carriers in shallow traps will dissipate first, while carriers in deep traps will be stored longer.

Figure R7. The **a)** electron and **b)** hole trap distribution of the Nylon and glass.

Here, we measured and calculated the carrier trap distribution of considered triboelectric materials based on the isothermal surface potential decay (ISPD) method. Figure R7 shows the obtained electron and hole trap distribution of Nylon and glass. It can be seen that the energy levels of electron and hole traps in Nylon are both below 0.9 eV. The electron shallow trap density (peak located at ~ 0.7 eV) is significantly higher than the deep trap (~ 0.8 eV), and there are almost no deep hole traps exist in Nylon. The electron and hole trap energy levels of glass are also below 0.85 eV, and the peak density of shallow traps (0.73 eV) is significantly higher than that of deep traps (0.8 eV). That is, shallow traps are the dominant carriers in Nylon and glass, indicating the weak charge storage capacity. On the contrary, the peak density of electron and hole traps in PTFE is around 1 eV, and the deep traps are dominant carriers. Therefore, Nylon and glass exhibit lower positive charge acceptance as most injected charges are stored in shallow traps, which will dissipate quickly in a short period. We have added relevant content to the revised manuscript. (Manuscript: Page 21, line 3-16, 23-29, Supporting Information: Supplementary Fig. 46-48)

6. In Figs. 4f and 4g, what causes the difference between a small peak of PTFE at ~ 0.8 eV and the large peak at ~ 1.0 eV?

Response: Thanks for your comments. Figures 5f and 5g (revised manuscript) give the electron and hole trap distribution of PTFE, PVC, Si rubber, and PE. The two peaks in the diagram represent the centers of shallow and deep traps, respectively. As mentioned above in question 5, deep traps have stronger charge binding or storage ability than shallow traps, and charge carriers in shallow traps will dissipate first. According to Figures 5f and 5g (revised manuscript), the shallow and deep trap centers of the electron and hole for PTFE are located at ~ 0.8 eV and ~ 1.0 eV, respectively. The peak intensity (area) of the deep trap is significantly larger than that of the shallow one. Therefore, it is difficult for the charge carriers to escape once they enter the PTFE traps, indicating the superior positive and negative charge storage and retention capabilities. We have added relevant content to the revised manuscript. (Manuscript: Page 21, line 23-29)

7. On page 21, the paragraph about a robotic e-skin is somewhat unclear. Please clarify why both positive- and negative-PTFE are used for e-skin and the research objectives of robotic experiments.

Response: Thanks for your suggestions. We have systematically confirmed that PTFE has excellent potential for customizing positive and negative surface charges, and the injected charges can be stably stored for a long time. To further highlight the application potential of PTFE with high surface charge density as triboelectric material, we designed and fabricated a single-electrode TENG with posi-PTFE and neg-PTFE as triboelectric layer and employed it as electronic skin for robot non-contact sensing. Generally, PTFE is widely regarded as a negative polarity triboelectric material, and we have confirmed that PTFE also has the ability to capture and stably store positive polarity charges in this work. We explored the output performance of neg-PTFE and posi-PTFE as triboelectric layers for non-contact TENG, respectively, as shown in Figure 6d-e (revised manuscript). As a proof of concept, we further employed posi-PTFE based TENG as electronic skin and evaluated the output performance in more depth to highlight the new application scenarios for positive polarity PTFE, as shown in Figure 6g-h (revised manuscript). We have added relevant content to the revised manuscript. (Manuscript: Page 26, line 1-4, 13-14)

8. Please describe the operational method of the robotic e-skin (module circuit, software, experimental condition, etc.) in the 'Methods' section.

Response: Thanks for your suggestions. In this work, a programmable manipulator (Hiwonder, uHandPi 2.0) was employed to conduct relevant demonstrations. Specifically, the five fingers and gimbal of the bionic manipulator are driven by the Hiwonder LFD-01 and LD-1501MG digital servo motor. The controller is based on an STM32 microcontroller. The PC control software (uHand V1.1) was developed based on the Windows Presentation Foundation (WPF). The experiment involved manipulating the displacement and action time of the digital servo motor connected to five fingers to accomplish various movements. For example, the finger action duration is set to 3s (gripping time: 1s, recovery time: 2s)

for non-contact perception, and the relative displacement ratio (corresponding to bending angle) is 178 (thumb), 208 (index finger), 282 (middle finger), 255 (ring finger), 144 (little finger) compared to the initial displacement of 1500. We have added relevant content to the revised manuscript. (Manuscript: Page 29, line 13-21)

Author's response to Reviewer 3

This manuscript chapter presents an approach to visualize and calculate surface charges in triboelectric nanogenerators using surface potential measurement. It links such measurements and charge extraction methods to enhancing performance in energy harvesting and self-powered sensing applications from triboelectric materials. The study introduces a surface charge imaging method using electrostatic surface potential measured by a Kelvin probe, coupled with an efficient tuning strategy for a single polarity surface charge based on corona discharge with a three-electrode design. This method allows visualization and quantification of surface charge distribution on triboelectric materials.

There are some innovative aspects to the research presented, like the mathematical algorithm to go from surface potential to surface charge.

The research also shows a 70-fold enhancement in output voltage for triboelectric polytetrafluoroethylene (PTFE), offering stable surface charge with minimal decay over time. Additionally, the charged PTFE is applied as a robot electronic skin for non-contact perception of object geometries, which is certainly an interesting application.

Some of these results will certainly be of interest and novelty to the scientific community in tribology and electrostatic nano-harvesters. However, I have serious concerns for the quality and representations given in this manuscript, and therefore I don't recommend for publication.

Response: Thanks for your valuable comments and careful review of this work. As you mentioned, the novelty and scientific value of this work for the community are concentrated in the following aspects.

First, we present a general strategy to visualize and quantify surface charge in triboelectric or dielectric materials by scanning the surface potential distribution combined with the mathematical inversion algorithm. This approach realizes the surface charge measurement for large-size samples with high surface potential value, which makes up for the limitations that *"the current Kelvin probe force microscopy methods (based on atomic force microscope) cannot measure a large sample scale of millimeters, nor can it handle a large value of surface potential > 10 V"* (as mentioned by Reviewer #1).

Second, we proposed an efficient tuning strategy for a single polarity surface charge based on corona discharge with a three-electrode design. The homogeneous and quantized injection of positive or negative charge on triboelectric materials was achieved with the help of the proposed visualization tool. The charge storage and dissipation mechanism for various common triboelectric materials were also analyzed, and the role of carrier traps on the charge behavior was clarified. We confirmed that the triboelectric material with high deep trap density could store contact electrification or deposited charges more durable by lowering the charge dissipation rate, which is significant for designing and fabricating high-performance TENGs.

Third, we found that the common tribo-negative material PTFE could be tuned as tribo-positive layer by injecting positive charges, delivering a 70-fold enhancement in output voltage for the identical

material-based TENG. As a proof of concept, the positive charged PTFE is also demonstrated as a robot electronic skin for non-contact perception of object geometries.

Additionally, we have thoroughly examined and comprehended your technical concerns regarding the following main points: 1) the algorithm accuracy and universality, 2) the injection charge stability mechanism. **Following your insightful comments, we added supplementary theoretical and experimental data to improve the manuscript, and tried our best to address your technical concerns as follows.** We would greatly appreciate if you could reconsider our revised manuscript.

-For example, while it is of use to have an accurate calculation of charge from surface potential, there is no detailed exploration of the accuracy of the method proposed against simulations.

Kelvin probe data can be very easily simulated in packages like COMSOL or ANSYS, allowing checking computational precision of the proposed maths for converting potential to charge.

This seems to the trust of the first part of the paper, but is purely a maths-filled jargon without any corroboration that the steps taken to improve accuracy indeed work, beyond the minimisation of error in the equations.

Response: Thanks for your comments. Actually, we have investigated the effects of noise levels on the charge inversion calculation results by manually setting the "TENG" letter-shaped charge in the original manuscript. We also evaluated the imaging quality of the charge distribution under various noise conditions utilizing "signal-to-noise ratio" (SNR) and "peak mean square error" (PMSE) as evaluation parameters. The image with high SNR and low PMSE indicates the high visualization quality. These parameters can reveal the accuracy of the proposed visualization and standardized quantitation (VSQ) method to a certain extent. However, as you mentioned, the accuracy of the proposed method against simulations still requires further clarification due to the lack of horizontal comparison. As a result, we have further validated this point in the revised manuscript.

Firstly, we supplemented a simulation example of "spotted-like" charge distribution (defined as *Case 1* in the revised manuscript). As shown in Figure R8a, the charge is distributed in a regular square pattern with charge density decreasing gradually from the center outwards, and the maximum charge density is set to $1 \text{ C}\cdot\text{m}^{-2}$. Then, we obtained the potential distribution under this charge density using COMSOL and applied diverse levels of noise interference to simulate the environmental disturbances, device circuit coupling, and other electronic components during the actual surface potential measurement process. Using this surface potential distribution as the input, we employed the proposed VSQ method to obtain charge inversion results under various noise levels, and the imaging quality of charge distribution was also evaluated by SNR and PMSE (as shown in Figure R8b). The results confirm that our proposed algorithm can obtain better charge distribution when the noise reaches 10%, with the SNR and PMSE of 28.24db and 3.87%, respectively.

Figure R8. The **a)** artificially set “spotted-like” and the induced surface potential distribution. **b)** The surface charge distribution under different Gaussian noise conditions obtained by the proposed visualization and standardized quantification (VSQ) method. **c)** The surface charge distribution under different Gaussian noise conditions obtained by the charge simulation (CS) method.

Figure R9. The **a)** artificially set charges (resembled the “TENG” letters) and the induced surface potential distribution. **b)** The surface charge distribution under different Gaussian noise conditions obtained by the proposed visualization and standardized quantification (VSQ) method. **c)** The surface

charge distribution under different Gaussian noise conditions obtained by the charge simulation (CS) method.

To further demonstrate the advantages and accuracy of our proposed VSQ method, we employed the "charge simulation" method based on the principle of direct inversion of surface potential distribution matrix to perform surface charge calculations on *Case 1* and *Case 2*, and the results are demonstrated in Figure R8c and Figure R9c. As for *Case 1* and *Case 2*, the image quality of charge distribution obtained by the CS method is lower than that of the proposed VSQ method (Figure R10). Moreover, the ill-posed of linear system constructed by Equation (4) becomes prominent under the high noise level conditions, resulting in a much lower SNR and a sharp increase in PMSE than our proposed algorithm. Therefore, our proposed VSQ method exhibits superior imaging quality compared to the CS method.

Figure R10. The SNR and PMSE of the proposed visualization and standardized quantification (VSQ) and charge simulation (CS) method when treated potential values with different noise levels. **a)** *Case 1*-the “spotted-like” charges. **b)** *Case 2*-the charges that resembled the “TENG” letters.

Figure R11. Comparison of the relative error of the visualization and standardized quantification (VSQ) and charge simulation (CS) method in *Case 1* and *Case 2*.

Furthermore, we introduced "relative error" to evaluate the quantitative accuracy, considering that the actual charge value in the simulation example is known. The relative error indicates the difference between the actual and estimated inversion calculation value, which can be defined as:

$$e = \frac{\|\sigma_{true} - \tilde{\sigma}\|_2}{\|\sigma_{true}\|_2} \quad (4)$$

where σ_{true} represents the actual value of the set surface charge, and $\tilde{\sigma}$ represents the estimated charge value. Figure R11 illustrates the relative error of the CS method and VSQ method for the two Cases under various noise levels. It can be observed that the relative error of charge inversion results gradually increases with the noise levels. The charge simulation method cannot achieve convincing results when the noise exceeds 2%. Specifically, the relative errors from the traditional CS method achieved 41.40% (*Case 1*) and 60.70% (*Case 2*) when the noise reached 10%, while the relative errors from our proposed VSQ method were only 13.36% (*Case 1*) and 15.46% (*Case 2*), respectively. This demonstrates that the proposed VSQ method demonstrates superior anti-interference capability, which is capable of restoring the charge distribution at higher noise levels.

Overall, the comparison of the relative error, SNR, and PMSE verifies that our proposed visualization and standardized quantification method approach exhibits exceptional resistance to interference and high accuracy in inversion. The content about error analysis has been added to the main text of the revised manuscript as well as the Supporting Information.

Besides, we noted that the reviewer mentioned that potential data obtained through the Kelvin probe can be imported into COMSOL or ANSYS, and the corresponding electrostatic module can be employed to calculate the surface charge distribution. In fact, the essence of charge calculation by COMSOL or ANSYS is based on the solution of the Poisson equation as well. Specifically,

The distribution characteristics of a space electric field can be described by the force F exerted on a unit charge q in the electric field, that is:

$$E = \lim_{q \rightarrow 0} \frac{F}{q} \quad (5)$$

It is apparent that the electric field intensity E is a vector function that varies with the position of the point in space, independent of the charge quantity. Considering Coulomb's law, the electric field intensity generated by a point charge (located at r') at point r can be expressed as follows,

$$E(r) = \frac{q(r - r')}{4\pi\epsilon_0 |r - r'|^3} \quad (6)$$

When there are multiple point charges in the space, the electric field at a point can be calculated by the vector sum of an electric field generated by each point charge at that point, that is,

$$E(r) = \sum_{i=1}^n E_i(r) = \frac{1}{4\pi\epsilon_0} \sum_{i=1}^n \frac{q(r-r')}{|r-r'|^3} \quad (7)$$

Thus, the electric field intensity generated by continuously distributed surface charges on surface S' can be represented as,

$$E(r) = \sum_{i=1}^n E_i(r) = \frac{1}{4\pi\epsilon_0} \int_{S'} \frac{\sigma(r')(r-r')}{|r-r'|^3} dS' \quad (8)$$

According to the irrotational nature of the electrostatic field, the relationship between the electric potential φ and the electric field E can be defined as,

$$E = -\nabla\varphi \quad (9)$$

Based on the definition of work done by an electric field on charge q , the electric potential at any point in the surface charge system can be solved:

$$\varphi(r) = \frac{1}{4\pi\epsilon_0} \int_{S'} \frac{\sigma(r')}{|r-r'|} dS' \quad (10)$$

Assuming that the angle between unit normal vector \mathbf{n} of an infinitesimal surface element dS' on the surface S' and the direction of electric field at that point is θ , and denoting $\mathbf{n}dS' = dS'$, then,

$$\mathbf{E} \cdot dS' = \frac{q}{4\pi\epsilon_0} \frac{(r-r')}{|r-r'|^3} \mathbf{n}dS' = \frac{q}{4\pi\epsilon_0} \frac{1}{|r-r'|^2} \cos\theta dS' \quad (11)$$

where $\mathbf{E} \cdot dS'$ represents the flux of electric field vector \mathbf{E} on the surface element dS' . By establishing a spherical coordinate system with the point charge position as the origin (i.e., $r'=0$), then,

$$\cos\theta dS' = |r-r'|^2 d\Omega \quad (12)$$

As a result,

$$\mathbf{E} \cdot dS' = \frac{q}{4\pi\epsilon_0} d\Omega \quad (13)$$

Integrating the above equation over the entire surface yields,

$$\oint_{S'} \mathbf{E} \cdot dS' = \begin{cases} \frac{q}{\epsilon} & \text{if } q \text{ lies inside } S' \\ 0 & \text{if } q \text{ lies outside } S' \end{cases} \quad (14)$$

Equation (14) is the integral form of the Poisson equation. It is evident that the linear system constructed in this manuscript (Equation (4) in the main text) is a specific form of the Poisson equation. The charge distribution can be obtained by directly inverting, which is also known as the apparent charge method or charge simulation method. This method exhibits excellent accuracy when the number of measured point n is small. However, the dimension of the transfer function matrix H rapidly increases with the increase of measured point n , bringing about an ill-conditioned problem of the linear system.

Even minor disturbances in observations can cause drastic changes in the solution of the equation, which may significantly impair its reliability. In this manuscript, we introduced iterative regularization, constructing a hybrid surface charge inversion algorithm based on Flexible Golub-Kahan iteration and Tikhonov regularization. Meanwhile, we adopted an adaptive weight generalized cross-validation (AWGCV) method to automatically select regularization parameters during the iteration process, thereby solving the problem of large inversion result errors due to noise interference.

Figure R12. **a**). i) The negative surface potential distribution of the charge spot induced by tip-plane electrode (sample size $20\text{mm} \times 20\text{mm}$). ii) The surface charge density obtained by COMSOL. iii) the surface charge density obtained by charge simulation (CS) method. iv) The surface charge density obtained by the proposed visualization and standardized quantification (VSQ) method. **b**). i) The positive surface potential distribution of the charge spot induced by tip-plane electrode (sample size $20\text{mm} \times 20\text{mm}$). ii) The surface charge density obtained by COMSOL. iii) The surface charge density obtained by charge simulation (CS) method. iv) The surface charge density obtained by the proposed visualization and standardized quantification (VSQ) method.

Figure R12 illustrates the surface charge distribution obtained by the CS method and COMSOL electrostatic module for the same measured electric potential distribution (surface charge was injected by tip-plate electrode). It is apparent that the calculation results of the CS method and COMSOL are consistent. However, the charge distribution edges are submerged owing to noise interference, preventing the acquisition of high-precision surface charge distribution. As a comparison, the image quality of charge distribution obtained by our proposed VSQ method (Figure R12a iv, Figure R12b iv)

is significantly better than that of COMSOL electrostatic module and the CS method (Figure R12a ii-iii, Figure R12b ii-iii).

Figure R13. The comparison of the surface charge distribution obtained by dust figure (powdering with printer toner particle) and the proposed VSQ method (scale bar: 3 mm).

To further verify the accuracy of our proposed VSQ method from an experimental perspective, we employed the “dust figure” method to provide a visual distribution result of the surface charge distribution as well ^[10]. This method is based on the principle of attraction between charges with opposite polarity, which blows charged dust particles into the space environment around the sample. The charged dust with different polarities will be deposited in charged regions of the sample, displaying the surface charge distribution. As shown in Figure R13, the charge distribution obtained by the VSQ method is consistent with the actual distribution measurement results based on the dust figure, verifying the accuracy of the algorithm proposed in this work. It should be noted that the dust figure is a destructive qualitative analysis method for surface charges, which cannot achieve quantitative analysis. Thus, our visualization and standardized quantification method demonstrate superior advantages.

In summary, we have systematically demonstrated the effectiveness of measures, including iterative regularization taken to improve the accuracy of charge inversion through experiments (dust figure) and simulations (COMSOL, charge simulation method), which effectively improves the quality of charge inversion images and reduce relative quantization errors. We have added relative content in the main text and Supporting Information in the revised manuscript. (Manuscript: Page 10-13, Supporting Information: Supplementary Fig. 1-5, Supplementary Note 1, 4-5)

-The stability of positive charges compared to negative ones in various environmental conditions is mentioned but not deeply analysed.

In fact, a large part of the literature in the field that already addresses this challenges has been entirely ignored. Several works link charge degradation to water presence on the materials, and lateral conduction, indicating that hydrophobic coatings can work as a solution to avoid charge decay. Here the authors merely changed the materials, but there is not real evidence of what is causing the degradation, and how a change in material improves it. See for example works in <https://doi.org/10.1109/IBERSENSOR.2014.6995520>, <https://doi.org/10.1109/94.660767>, <https://doi.org/10.1109/PVSC.2014.6924985>

Response: Thanks for your comments. In order to investigate surface charge stability, we first explored the charge dissipation properties of typical triboelectric materials, such as PTFE, PFA, PVC, PP, PET, PE, Si rubber, glass and Nylon after being charged with both positive and negative charges. It was discovered that PE, silicone rubber, Nylon, and glass are nearly incapable of storing charges for an extended period, while PTFE and PFA demonstrate far greater charge storage capacities than the other materials.

Figure R14. Schematic diagram of the distribution of shallow and deep traps for electrons and holes in tribo-dielectric material.

In order to explain the underlying reasons for the differences in charge storage capacity among the above-mentioned materials, we further tested their carrier traps (shadow trap, deep trap) distribution. Specifically, carrier traps in dielectric material belong to localized states with energy levels in the bandgap. Carrier traps are divided into two types: deep traps and shallow traps. The schematic diagram of their distribution in the energy band structure is shown in Figure R14. Electron traps (electron/negative ion capture sites) and hole traps (positive ion capture sites) are distributed on both sides of the Fermi level. Using hole traps as an illustration, the shallow traps are closer to the bottom of the valence band, indicating that the energy required for the carrier in the shallow trap is low. However, the deep traps are situated away from the valence band and near the Fermi level, and the energy required

for the carrier to escape from the deep trap is high. The positive/negative charges injected into the dielectric material through corona discharge are captured by both the shallow and deep traps, and the stored charges in the traps will undergo dissipation when the applied voltage is removed. In general, charge carriers in shallow traps will dissipate first, while carriers in deep traps will be stored longer.

Figure R15. The electron trap level distribution of the common triboelectric materials.

Figure R16. The hole trap level distribution of the common triboelectric materials.

Isothermal surface potential decay (ISPD) is the most widely used method for characterizing carrier traps in dielectric materials. Herein, we measured the carrier trap distribution of considered triboelectric materials based on the ISPD method. Figure R15-16 illustrates that materials with great charge storage capacities (slower charge dissipation rate), such as PTFE, PFA, and PVC, have far larger densities of deep traps (holes and electrons) than shallow traps. For instance, PTFE possesses nearly entirely a high-energy level deep trap located at 1.0 eV. As for PE, the densities of deep and shallow traps are similar. Triboelectric materials with lower charge storage capacities, such as glass, Si rubber, and Nylon, have much higher densities of shallow traps than deep traps. That is, shallow traps are dominant carrier traps. Therefore, we have conducted carrier trap distribution analysis to explain the difference in positive and negative charge storage capacity of the considered triboelectric materials. It can be concluded that the intrinsic carrier trap distribution of the materials essentially determines the charge storage and dissipation properties. Improving the energy level and density of deep traps in dielectric materials is the key to optimizing their surface charge storage capacity, which is of great value for developing high-performance triboelectric materials and devices. At the same time, we also noted that the reference provided (Cao et al. "Study of porous dielectrics as electret materials." *IEEE Transactions on Dielectrics and Electrical Insulation* 5.1 (1998): 58-62.) also utilized the ISPD method to characterize and explain the reasons for charge dissipation and the role of carrier traps were also mentioned ^[11]. Mizsei et al. also pointed out that "The microscopic charge is surprisingly stable on the dry surface. In spite of the huge lateral ($\sim 700,000$ V/m) and vertical (~ 0.25 V/2.5 nm = 10^8 V/m) electric field, the charge distribution hardly changes within 20 h. It is not easy to explain this stability. Probably traps exist on the oxide surface, thus the surface mobility of the charges is very low." ^[12] Therefore, using trap distribution characteristics to explain the differences in charge storage capacity of dielectric materials is a universal and reasonable method.

Figure R17. Mechanism of the surface charge dissipation on polymer surface.

Furthermore, we also examined the primary channels and mechanisms of surface charge dissipation, 1) dissipation through the volume of the dielectric material (bulk conductivity), 2) dissipation along the surface of the dielectric material (surface conductivity), or 3) neutralization with charged particles in the gas or air (as depicted in Figure R17). Typically, these three pathways exist simultaneously. When bulk conductivity is dominated, the surface charge distribution will not change and the dissipation rate will essentially be constant across the board. We discovered that the surface positive and negative charge densities of materials with larger bulk conductivity (such as silicone rubber) decrease uniformly during dissipation, indicating that bulk dissipation is the dominant pathway. The surface charge distribution morphology will change when pathways 2) and 3) are the primary for dissipation. According to the test results, the negative surface charge was evenly distributed throughout a month (4 weeks) for PTFE, PFA, PVC, PET, PP, and no surface charge diffusion was found towards the surrounding area. The charge density at the edge of the sample slightly decreases after 6 weeks of dissipation, while the overall charge distribution remains uniform. Regarding positive charges, a slight decrease in the PFA, PVC, PET, and PP surface charge density around the edge area was found after 3 days, indicating that the contribution of surface conductivity in the positive charge dissipation process is slightly larger than that of negative charges. In general, the surface charge stability of PTFE, PFA, PVC, PET, and PP is excellent, and surface conduction is not the primary dissipation channel as these insulation materials possess low surface conductivity.

In addition, we also explored the influence of relative humidity on the surface charge dissipation of PTFE in the original manuscript. We carefully read the relevant references provided by you and found that hexamethyldisilazane (HMDS) was used as the interface modification. It was pointed out that HMDS provides a highly hydrophobic monolayer coating that prevents water from being absorbed. In fact, *Tschentscher* et al. presented a study on the influence of humidity on the low-field conduction processes and found that the ionization process of gases is unaffected by the low electric field produced by surface charges, where they pointed out that “*No influence of humidity was detected on the ion-pair generation from natural ionisation in the gas*” and “*the low-field conduction phenomena did not significantly change with the gas humidity*”^[13]. Under high humidity conditions, water molecules will create a thin layer of water film on the dielectric's surface. This will lead to a higher involvement of surface charge conduction during dissipation, and the distribution of surface charge will expand in the tangential direction. Here, we compared the positive and negative surface charge distribution of PTFE under 30% R.H., 90% R.H. conditions, as shown Figure R18-19. It can be found that the surface charge density is uniformly distributed and there are no discernible edge diffusion phenomena after 4 weeks in the low-humidity environment. However, both positive and negative surface charge density demonstrate a significant diffusion in the surrounding edge areas after two hours of exposure to the 90% R.H. environment. The surface charge distribution becomes less uniform as the dissipation time increases, resulting in an uneven distribution after 4 weeks of exposure. Actually, we noticed that you also

mentioned that “Several works link charge degradation to water presence on the materials, and lateral conduction, indicating that hydrophobic coatings can work as a solution to avoid charge decay”. These conclusions are consistent with the relevant test results in this work. That is, the presence of a water film leads to an increase in surface conductivity, accelerating the surface charge dissipation rate under high-humidity environment. This process may be efficiently inhibited by constructing hydrophobic coatings to prevent charge decay.

Figure R18. The 3D negative surface charge distribution of the PTFE during long-term dissipation. **a)** 30% R.H. **b)** 90% R.H.

Figure R19. The 3D positive surface charge distribution of the PTFE during long-term dissipation. **a)** 30% R.H. **b)** 90% R.H.

It should be noted that increasing the humidity resistance of surface charges by surface modification will not alter or modify the dissipation pathways and mechanisms, and it essentially suppresses the charge dissipation pathways caused by humidity. Restricted by the length of this manuscript, we have not conducted associated surface modification research. Here, we aim to introduce the surface charge

visualization and standardized quantitation method, which was further employed to assess and describe the surface charge polarity control, storage, and dissipation of common triboelectric materials. The validity and universality of the proposed strategies have been highlighted, and several valid conclusions were obtained. We have enhanced the analysis of charge stability mechanism and updated relevant content in the main text and Supporting Information in the revised manuscript. (Manuscript: Page 21-23, Supporting Information: Supplementary Fig. 46-48, 51-52)

-Some aspects do not seem to be immediately relevant and sufficiently supported:

-Line 28, why is the visualisation and standardisation of surface charges essential for high-performance nanogenerators? – it is clearly important, but certainly not a prerequisite. In fact, the text itself later on described the charge density as the primary requirement for improved performance.

Response: Thanks for your comments. We mentioned that visualization and standardized quantification of surface charges are essential for developing high-performance TENG. Undoubtedly, the surface charge density is the primary determinant of TENG's output characteristics. Achieving high performance requires the increase of surface charge density and minimizing the adverse effects of charge dissipation in triboelectric materials. However, due to the lack of macroscopic visualization and standardized quantification tools for surface charge density, it is currently challenging to establish the correlation between device output and the surface charge density of triboelectric materials. In this work, we proposed a universal approach and tool to solve this problem by combining the scanned surface potential with high-precision charge inversion algorithms. This work is crucial for an intuitive and quantitative understanding of the transfer mechanism of contact electrification, large-scale interface charge distribution and evolution characteristics, and guiding the development of high-performance TENG materials and devices.

For example, we can achieve a deeper understanding of the triboelectric charge behavior during contact electrification under different working conditions, as shown in Figure 3b-c (revised manuscript). Meanwhile, surface charge visualization can also be used to track the aberrant states on the triboelectric layer, such as the surface "defect", the presence of nonuniform contact separation during TENG operation. As a proof of concept, we demonstrated the recognition of depressions in the triboelectric layer by mapping the surface charge density, as shown in Figure 6a (revised manuscript). The standardized quantification of surface charge can accurately calculate the surface charge density of the triboelectric layer under different conditions, which helps to establish the connection with the actual electrical output and provides guidance for developing high-performance triboelectric materials and devices. We have further strengthened the relevant explanations in the revised manuscript. (Manuscript: Page 3, line 2-5)

-Line 82: "However, the ion deposition process is random and cannot be visualised or standardised. The needle-plate electrode": This is incorrect, as can be seen from work in references

doi.org/10.1016/j.apsusc.2017.03.204,

doi.org/10.1016/j.apsusc.2006.03.075,

doi.org/10.1016/j.polymdegradstab.2014.03.017, doi.org/10.1016/j.egypro.2016.07.090

Response: Thanks for your comments. Here, we aim to express that the accurate charge deposition into the dielectric through corona discharge by traditional point-to-plane electrodes is challenging. Although the corona discharge parameters, including voltage, pulse current, and duration time, could be controlled, the amount and distribution of deposited charges at the dielectric surface cannot be predicted accurately. This is also mentioned by *János Mizsei* in their work (<https://doi.org/10.1016/j.apsusc.2006.03.075>), *“One advantage of the corona charge is that it is a contact free procedure. As it is discussed above, the surface charge can be measured after the corona charging process, but the surface potential cannot be estimated exactly in advance.”* They also mentioned that *“The deposited charge can be measured by integrating the charging current during the charging process, and independently by vibrating capacitor over the surface, if the insulator properties are well known”*. In this work, we proposed a strategy for surface charge visualization and standardized quantification, which is capable of addressing the needs in the field.

We have modified the original expression into “It is challenging to realize accurate surface charge deposition through corona discharge by traditional tip-plane electrodes” in the revised manuscript. Besides, the “needle-plate electrode” was also changed to “tip-plane” according to relevant references. (Manuscript: Page 4, line 2-5, Page 15, line 12-17)

-What the authors refer to as standardization is entirely unclear. It appears they are misusing the word, or have a different meaning intended.

Response: Thanks for your comments. Here, standardization means a universal method with high accuracy for quantifying surface charges of triboelectric materials. As an essential tool, this method can provide guidance for surface charge-related fields, including contact electrification mechanisms and the development of triboelectric materials. For example, design a material with tailorable triboelectric properties that could meet desired device requirements. In order to highlight the universality of this method, we have chosen the word "Standardization". We selected this expression in the hope that more efforts can be dedicated to standardizing the measurement of surface charge in triboelectric materials and deeply exploring the intrinsic relationship between surface charge and TENG output. Thank you for your suggestion. We have changed “standardization” to “standardized quantification” in the algorithm part of the revised manuscript to highlight its significant value, and replace this word in some parts to avoid ambiguity. Besides, we also changed the title of this manuscript to “Visualization and Standardized Quantification of Surface Charge Density for Triboelectric Materials” in the revised version.

-A large part of the argument is based around triboelectric materials, but all experiments are conducted using corona discharge, which uses a different mechanistic approach to building charge concentration on a surface.

Response: Thanks for your comments. The performance of TENG is primarily governed by the ability of the triboelectric materials to generate, store, and dissipate surface charge, as well as their surface charge density. In this study, we have introduced a novel method for visualizing and quantifying the surface charge of triboelectric material, aiming to overcome the limitations of existing methodologies. On this basis, we explored the charge storage and dissipation characteristics of various common triboelectric materials (PTFE, PFA, PVC, PP, PET, PE, Si rubber, Nylon, glass) in the triboelectric series. Subsequently, the "three-electrode" corona discharge approach is intended to achieve uniform positive or negative charge injection or deposition on triboelectric materials, and the relationship between surface charge density and TENG output was explored. It has been proven that external positive charge injection can reverse the tribo-polarity of PTFE, the most widely used tribo-negative material. The output of identical material based TENG can also be significantly improved through surface charge density control.

Moreover, we examined the positive polarity charge storage and dissipation properties of typical triboelectric materials. We further analyzed the distributions of deep, shallow charge carrier traps and elucidated the crucial function of deep carrier traps for long-duration charge storage. In other words, materials with a larger density of deep traps exhibit greater capacity for storing surface charge, which offers valuable insights and direction for the advancement of triboelectric materials and devices with enhanced performance.

In summary, this work utilizes the single polarity charge created by corona discharge to reveal the relationship between surface charge density and triboelectric output, and elucidates the significance of deep carrier traps in triboelectric materials. The above research and significant conclusions are based on using corona discharge to inject charges and establish different surface charge densities. This is also the most direct demonstration of the surface charge visualization and standardized quantification strategy proposed in this work. In the revised manuscript, we further explained and clarified the correlation between the corona discharge used and the surface charge behavior of triboelectric materials. (Manuscript: Page 15, line 12-17, Page 18, line 8-9)

-Line 183: "Accordingly, we proposed the adaptive weight generalized cross validation (AWGCV) to select the regularization parameter η , as illustrated in Fig. 1c iii." – This figure is merely showing a set of equations, which are already described in the text, to exemplify the iteration flow of finding a right approach to solving the linear system, but it does not say anything about why, how, and the advantages to using AWGCV.

Response: Thanks for your comments. After obtaining the relationship between electric potential and charge distribution (Equation (4)), the charge visualization method employed in this work projected equation (4) onto a lower-dimensional Krylov subspace through Flexible Golub-Kahan decomposition, and then solved the resulting projected least squares problem through Tikhonov regularization. This method provides a natural framework to efficiently avoid nested loops of iterations by regarding the

inverse of the regularization matrix (stemming from an iteratively reweighted regularization term) as iteration-dependent pre-conditioning in $\min\{\|\mathbf{H}\boldsymbol{\sigma} - \boldsymbol{\varphi}\|^2 + \lambda^2 \mathbf{I} \|\boldsymbol{\sigma}\|^2\}$. During this process, we employed the AWGCV method to select the regularization parameters at each iteration, improving the problem of GCV selecting overly large regularization parameters in projected least squares problems, which can result in excessively smooth solutions.

GCV method can find appropriate regularization parameters even when the noise level is unknown. This approach is derived from the concept of cross-validation, in which the original data is split into two portions: one portion is utilized to calculate an approximate solution, while the other portion is employed to validate this approximation. For the standard Tikhonov regularization problem ($\min\{\|\mathbf{H}\boldsymbol{\sigma} - \boldsymbol{\varphi}\|^2 + \lambda^2 \mathbf{I} \|\boldsymbol{\sigma}\|^2\}$), by sequentially removing individual data points from the measurement matrix $\boldsymbol{\varphi}$, the GCV function below is used to find regularization parameters that minimize the prediction error:

$$G(\lambda) = \frac{n \left\| (\mathbf{I} - \mathbf{H}\mathbf{H}_\lambda^\dagger) \boldsymbol{\varphi} \right\|_2}{(\text{trace}(\mathbf{I} - \mathbf{H}\mathbf{H}_\lambda^\dagger))^2} \quad (15)$$

where $\text{trace}(\cdot)$ represents the trace of a matrix, and $\mathbf{H}_\lambda^\dagger$ represents the pseudo-inverse of matrix $[\mathbf{H}, \lambda \mathbf{I}]^T$. As for the standard Tikhonov regularization problem, GCV is a method with high accuracy for selecting regularization parameters when the noise is a Gaussian signal. However, the GCV function may exhibit regions of excessive flatness near local minima for certain specific problems, which can lead to overestimation. Overall, the GCV can provide good computational accuracy.

However, when using the GCV method to select regularization parameters for the projected regularization problem, the solution error cannot stabilize well as the iteration progresses. Existing research has shown that in each iteration of the projected least squares problem, regularization parameters selected by the standard GCV function tend to be too large, leading to poor convergence behavior. This may be attributed to round errors introduced during the computation of \mathbf{Z}_k , \mathbf{W}_k , and \mathbf{D}_k during the flexible Golub-Kahan decomposition. Therefore, we introduced the concept of Weighted GCV (WGCV) in parameter selection. As for the standard Tikhonov regularization problem ($\min\{\|\mathbf{H}\boldsymbol{\sigma} - \boldsymbol{\varphi}\|^2 + \lambda^2 \mathbf{I} \|\boldsymbol{\sigma}\|^2\}$), WGCV can be expressed as:

$$G(\omega, \lambda) = \frac{n \left\| (\mathbf{I} - \omega \mathbf{H}\mathbf{H}_\lambda^\dagger) \boldsymbol{\varphi} \right\|_2}{(\text{trace}(\mathbf{I} - \omega \mathbf{H}\mathbf{H}_\lambda^\dagger))^2} \quad (16)$$

Therefore, for the projected least squares problem $\min\{\|\mathbf{D}_k \mathbf{y}_k - \beta \mathbf{e}_1\|^2 + \eta \|\mathbf{y}_k\|^2\}$ obtained through flexible Golub-Kahan decomposition, its WGCV can be expressed as:

$$G(\omega, \eta) = \frac{k \|(I - D_k D_{k,\eta}^\dagger) \beta \mathbf{e}_1\|_2}{(\text{trace}(I - \omega D_k D_{k,\eta}^\dagger))^2} \quad (17)$$

When $\omega=1$, the above equation degenerates into standard GCV; if $\omega>1$, an over-smoothed solution is obtained; if $\omega<1$, the obtained solution is under-smoothed. At present, researchers tend to employ experimental methods to select weight parameters. The specific approach involves calculating the solution for the k -th iteration under different ω values, given a set of accurate solutions σ_{true} . The optimal weight parameters can be determined when the relative error is minimized. However, the noise and randomness in measurement data can make the experimentally chosen weight parameters unsuitable for all situations in practical application, and they involve significant subjective elements. Therefore, an adaptive WGCV (A-WGCV) method is employed in this work, which automatically selects regularization parameters for the projected least squares problem and addresses the limitations of selecting experimental parameters.

Define the singular value decomposition (SVD) of matrix D_k as,

$$D_k = P \begin{bmatrix} A \\ \mathbf{0}^T \end{bmatrix} Q^* \quad (18)$$

where P is a unitary matrix of order $(k+1) \times (k+1)$; $A = \text{diag}(\delta_1, \delta_2, \dots, \delta_k)$ is a diagonal matrix of order $k \times k$, and the elements of its diagonal are the singular values of the matrix D_k that arranged in the order of $\delta_1 \geq \delta_2 \geq \dots \geq \delta_k \geq 0$; Q^* is the conjugate transpose of Q , which is a unitary matrix of order $k \times k$. The regularization solution of $\min \left\{ \|D_k \mathbf{y}_k - \beta \mathbf{e}_1\|^2 + \eta \|\mathbf{y}_k\|^2 \right\}$ is equivalent to:

$$\mathbf{y}_k = \sum_{i=1}^k \psi_i \frac{p_i^T \beta \mathbf{e}_1}{\delta_i} q_i \quad (19)$$

where, p_i and q_i are the elements in the matrices P and Q , respectively, $\psi_i = \frac{\delta_i^2}{\delta_i^2 + \eta^2}$ represents the Tikhonov filter factor with its range in interval $[0, 1]$.

Using the relation of (18) and (19), the WGCV function is transformed into:

$$G(\omega, \eta) = \frac{k\beta^2 \left(\sum_{i=1}^k \frac{\eta^2}{\delta_i^2 + \eta^2} p_i^T \mathbf{e}_1 \right)^2 + (p_{k+1}^T \mathbf{e}_1)^2}{\left(1 + \sum_{i=1}^k \frac{(1-\omega)\delta_i^2 + \eta^2}{\delta_i^2 + \eta^2} \right)^2} \quad (20)$$

Suppose $\eta_{k, \text{opt}}$ is the optimal regularization parameter of the k^{th} iteration. Since FGK is not sensitive to ill-posed problems in early iterations, we can assume that $\eta_{k, \text{opt}}$ satisfies,

$$0 \leq \eta_{k, \text{opt}} \leq \delta_{\min}(D_k) \quad (21)$$

where $\delta_{\min}(D_k)$ is the minimum singular value of matrix D_k in each iteration. Assuming that $\eta_{k, \text{opt}}$ is known, the ω can be found by minimizing the GCV function by the partial derivative with respect to η

from (20), namely,

$$\frac{\partial}{\partial \eta}(G(\omega, \eta)) \Big|_{\eta=\eta_{k, opt}} = 0 \quad (22)$$

Since $\eta_{k, opt}$ is unknown, we instead find ω corresponding to $\eta_{k, opt} = \delta_{\min}(\mathbf{D}_k)$. In later iterations, this approach fails because $\delta_{\min}(\mathbf{D}_k)$ becomes nearly zero due to ill-conditioning. Thus, in order to prevent ω from being too small in subsequent iterations, the ω_k at k th iterations is the average value in the previous iterations, i.e.

$$\omega_k = \text{mean}\{\omega_1, \omega_2, \dots, \omega_{k-1}\} \quad (23)$$

The above discussion is the theoretical derivation of the AWGCV method and principles for selecting the weight parameter ω . The adjustment of regularization parameters through the weight parameter ω can be employed to achieve better convergence characteristics and more accurate solutions.

Figure R20. **a)** The surface charge distribution under different Gaussian noise conditions obtained by the proposed method treated with GCV approach for *Case 1*. **b)** The surface charge distribution under different Gaussian noise conditions obtained by the proposed method treated with GCV approach for *Case 2*.

To further illustrate the necessity and accuracy of the AWGCV method, we applied the GCV method to calculate charge inversion results for *Case 1* and *Case 2* based on the VSQ method, as shown in Figure R20. Similarly, the SNR and PMSR were used to evaluate the charge imaging quality of the GCV method, and relative error was used to measure the difference between the estimated values and the actual values. Regarding image quality, *Case 1* and *Case 2* both exhibit considerable inversion capability with the GCV method under low noise conditions. However, as the noise increases, the SNR of the GCV method decreases compared to the AWGCV method, while PMSR increases (as shown in Figure R21). Furthermore, at high noise levels (5% and 10%), the GCV method exhibits the phenomenon of over-smoothing, leading to the maximum charge value much smaller than $1 \text{ C}\cdot\text{m}^{-2}$ (Figure R22). The relative error also follows the same trend, with the accuracy of the GCV method lower than that of AWGCV method under high noise conditions.

Figure R21. The SNR and PMSE of the AWGCV and GCV approach when treated with potential values under different noise levels. **a)** *Case 1*. **b)** *Case 2*.

Figure R22. **a)** Comparison of the relative error of the GCV and AWGCV approach for *Case 1*. **b)** Comparison of the relative error of the GCV and AWGCV approach for *Case 2*.

In conclusion, the AWGCV method exhibits better stability and inversion accuracy, particularly overcoming the drawbacks of the GCV method, where the selection of too large regularization parameters will lead to reduced accuracy, especially in high-noise conditions. This makes the AWGCV method highly advantageous in charge imaging or visualization. We have incorporated these findings into the revised manuscript. (Manuscript: Page 12, line 4-13, Supporting Information: Supplementary Fig. 4-5)

-Also, another crucial questions have not been addressed:

If the distance between the probe and the sample changes (as it would be the case in Kelvin probe) the transfer matrix H in their algorithm would change. So does this render the algorithm only useful if using exactly the same machine with the same parameters?

Response: Thanks for your comments. **The change in probe-to-surface distance will only impact the precision of surface potential measurement and have any influence on the universality of the proposed charge visualization and standardized quantification method.**

In this work, the Kelvin electrostatic probe was employed to scan the sample's surface potential distribution. The sample surface was divided into grids with n elements. Then, the surface potential and surface charge distribution were expressed in the following matrix form, where $H_{n \times n}$ is defined as the transfer function matrix for surface charge inversion calculation.

$$\begin{bmatrix} \varphi_1 \\ \vdots \\ \varphi_i \\ \vdots \\ \varphi_n \end{bmatrix}_{n \times 1} = \begin{bmatrix} h_{11} & \cdots & h_{1j} & \cdots & h_{1n} \\ \vdots & \ddots & \vdots & \ddots & \vdots \\ h_{i1} & \cdots & h_{ij} & \cdots & h_{in} \\ \vdots & \ddots & \vdots & \ddots & \vdots \\ h_{n1} & \cdots & h_{nj} & \cdots & h_{nn} \end{bmatrix}_{n \times n} \begin{bmatrix} \sigma_1 \\ \vdots \\ \sigma_i \\ \vdots \\ \sigma_n \end{bmatrix}_{n \times 1} \quad (24)$$

$$\mathbf{H}_{n \times n} = \begin{bmatrix} h_{11} & \cdots & h_{1j} & \cdots & h_{1n} \\ \vdots & \ddots & \vdots & \ddots & \vdots \\ h_{i1} & \cdots & h_{ij} & \cdots & h_{in} \\ \vdots & \ddots & \vdots & \ddots & \vdots \\ h_{n1} & \cdots & h_{nj} & \cdots & h_{nn} \end{bmatrix}_{n \times n} \quad (25)$$

In response to the first question, we provided the basic logic for constructing the H matrix, which is based on the Poisson equation of the electrostatic field. The dielectric material's surface is split into n grid components using the finite element method after being scanned with a Kelvin electrostatic probe to determine its potential distribution. The charges and potentials within the grid are uniformly distributed when the grid is small enough. According to the derivation of the Poisson equation, the potential generated by point j at point i can be expressed as,

$$\varphi(i) = \frac{1}{4\pi\epsilon_0} \sum_{j=1}^n (\sigma)_j \int_{S_j} \frac{1}{r_{ij}} dS \quad (26)$$

where S_j is the area of the j^{th} element, r_{ij} is the distance from point i to point j . Equation (1) states the potential-charge connection in matrix form for every grid covering the whole area. The elements of the H matrix may be expressed as follows,

$$h_{ij} = \frac{1}{4\pi\epsilon_0} \int_{S_j} \frac{1}{r_{ij}} dS \quad (27)$$

It is apparent that the H matrix is independent of the distance between the electrostatic probe and the sample, which solely relates to the distance between points i and j , or the form of the tested sample and the placement of the ground electrode. That is to say, there are no limits for the devices (equipment) and parameter settings for potential measurement.

TREK ELECTROSTATIC VOLTMETER 341B

TECHNICAL DATA	
Performance Specifications	
Measurement Range	0 to ±20 kVDC or peak AC
Measurement Accuracy	Better than ±0.1% of full scale, referred to the voltage monitor
Speed of Response	Less than 200 μs for 1 kV step. Less than 5 ms for 20 kV step change (10 to 90%)
Full Signal Bandwidth	DC to better than 25 Hz
Stability	Drift with Time: Less than 100 ppm/hour, noncumulative Drift with Temperature: Less than 100 ppm/°C
Operation Conditions	
Temperature	0 to 40°C (32 to 104°F)
Relative Humidity	To 90%, noncondensing
Altitude	To 2000 m (6561.68 ft)
Probe-to-Surface Separation	3 mm ±1 mm (recommended)

Figure R23. The technical parameters of electrostatic potentiometer (Terk 341B).

Moreover, the measurement precision of the surface potential is determined by the operating principle of the electrostatic probe, and this accuracy can be affected by variations in the distance between the probe and the sample. In this work, the electrostatic potentiometer (TREK 341B) is used, which has a maximum measurement error of ± 0.1% and a measurement range of -19.99 kV to +19.99 kV. The measurement data is transmitted to the digital oscilloscope through the BNC output port (with an output ratio of 1000:1). The matching probe used in this work is the TREK 3455ET, with a working response speed of less than 200 μs for a 1 kV step. According to the equipment handbook, the suggested distance between the probe and the sample is 3 mm ± 1 mm (2-4 mm) (Figure R23). Additionally, we examined the surface potential measurement results at various probe-to-sample distances (1–15 mm) (Figure R24) and discovered that the average surface potential (-1570~-1572V) was observed at a distance of 1-4 mm. The surface potential measurement findings will gradually be underestimated as the distance increases to higher than 5 mm. In particular, the test results will be significantly distorted as the distance exceeds 9 mm. Therefore, changes in the probe-to-sample distance only impact the precision of surface potential distribution measurement and have no impact on the charge inversion calculation

process. In this paper, we ensure that the distance between the probe and the sample to be tested is 3 mm, which is able to meet the requirements of the probe response to voltage.

Figure R24. The influence of probe-to-surface distance (d) on the surface potential measurement accuracy. a) The relationship between detected surface potential and d . b) The 2D surface potential distribution of the 15×15 mm PTFE sample under different probe-to-surface distances.

Figure R25. The a) structure, b) surface potential and c) surface charge distribution of the 3D truncated cone sample obtained by the proposed charge visualization and standardized quantification method.

Additionally, researchers can solve the H matrix for charge inversion in accordance with unique test criteria by utilizing the strategy presented in this work. As a proof of concept, we demonstrated the application of this method to three-dimensional surface charge inversion. Specifically, we created a truncated cone-shaped sample, and a circular surface potential scanning route was used to determine its potential distribution. We further achieved the visualization and standardized quantification of the surface charge distribution by inversion, as shown in Figure R25.

Overall, the VSQ method proposed in this work is applicable to two-dimensional, three-dimensional sample architectures and does not impose any particular constraints on the devices or parameters used for surface potential measurement. We have added relevant content to the revised manuscript. (Manuscript: Page 13, line 8-19, Supporting Information: Supplementary Note 6)

-It is unclear what Fig 1.a (first image) is supposed to be communicating. Not covered in the text, but it also just seems like a gimmick rather than a scientific schematic of relevance.

Response: Thanks for your comments. Figure 1a aims to illustrate the correlation between the surface potential and surface charge during TENG operation. Specifically, TENG based on the principles of contact electrification and electrostatic induction, generates triboelectric charges and surface potential (electric field) on the triboelectric material surface during the repetitive contact-separation process. In the original Figure 1a, the potential and surface charge distribution of the triboelectric layer with letter "T" pattern was presented, which is somewhat misleading. We have redrawn Figure 1a (as shown in Figure 33) in the revised manuscript and further explained the inherent correlation between TENG, surface charge, and potential.

Figure R26. The correlation between surface potential and surface charge density for TENG. Triboelectric surface charge generated during the repetitive contact-separation process of TENG will form surface potential that affects the electrostatic induction performance. (Manuscript: Page 6, line 9-10)

-The units and magnitudes in the colour bars are not readable.

Response: Thanks for your comments. We have enlarged the units and magnitudes in the colour bars in the revised manuscript and Supporting Information. (Manuscript: Pages 1-6, Supporting Information: Supplementary Figs. 1-2, 4, 6-7, 9, 11-14, 16, 18-19, 22-45, 49-53)

-The images that portray real data do not have a lateral scale bar.

Response: Thanks for your suggestions. We have added the lateral scale bars in all the relevant graphs in the revised manuscript and Supporting Information. (Manuscript: Pages 1-6, Supporting Information: Supplementary Figs. 1-2, 4, 6-7, 9, 11-14, 16, 18-19, 22-45, 49-53)

-Finally, the level of scientific English in this article is also unsatisfactory:

- Line 26 is grammatically incorrect. Missing an article.

- Line 52

- Line 53, by a direct visually.

- And many others.

Response: Thanks for your suggestions. We have checked the whole manuscript carefully and corrected relevant grammar and typing errors in the revised manuscript. All the changes are marked in *red characters*. Thanks again for your careful review.

References

- [1] Li, Z., Zhu, M., Shen, J., Qiu, Q., Yu, J., & Ding, B. (2020). All-fiber structured electronic skin with high elasticity and breathability. *Advanced Functional Materials*, 30(6), 1908411.
- [2] Lu, D., Liu, T., Meng, X., Luo, B., Yuan, J., Liu, Y., ... & Nie, S. (2023). Wearable triboelectric visual sensors for tactile perception. *Advanced Materials*, 35(7), 2209117.
- [3] Ning, C., Dong, K., Cheng, R., Yi, J., Ye, C., Peng, X., ... & Wang, Z. L. (2021). Flexible and stretchable fiber-shaped triboelectric nanogenerators for biomechanical monitoring and human-interactive sensing. *Advanced Functional Materials*, 31(4), 2006679.
- [4] Shrestha, K., Pradhan, G. B., Asaduzzaman, M., Reza, M. S., Bhatta, T., Kim, H., ... & Park, J. Y. A Breathable, reliable, and flexible siloxene incorporated porous SEBS-based triboelectric nanogenerator for human-machine interactions. *Advanced Energy Materials*, 2302471.
- [5] Shi, Y., Yang, P., Lei, R., Liu, Z., Dong, X., Tao, X., ... & Chen, X. (2023). Eye tracking and eye expression decoding based on transparent, flexible and ultra-persistent electrostatic interface. *Nature Communications*, 14(1), 3315.
- [6] Fu, X., Pan, X., Liu, Y., Li, J., Zhang, Z., Liu, H., & Gao, M. Non-contact triboelectric nanogenerator. *Advanced Functional Materials*, 2306749.
- [7] Apodaca, M. M., Wesson, P. J., Bishop, K. J., Ratner, M. A., & Grzybowski, B. A. (2010). Contact electrification between identical materials. *Angewandte Chemie International Edition*, 49(5), 946-949.
- [8] Xu, C., Zhang, B., Wang, A. C., Zou, H., Liu, G., Ding, W., ... & Wang, Z. L. (2019). Contact-electrification between two identical materials: curvature effect. *ACS nano*, 13(2), 2034-2041.
- [9] Wang, Z. L. (2021). From contact electrification to triboelectric nanogenerators. *Reports on Progress in Physics*, 84(9), 096502.
- [10] Li, C., Zhu, Y., Zhi, Q., Sun, J., Song, S., Connelly, L., ... & Mazzanti, G. (2021). Dust figures as a way for mapping surface charge distribution-A review. *IEEE Transactions on Dielectrics and Electrical Insulation*, 28(3), 853-863.
- [11] Cao, Y., Xia, Z., Li, Q., Shen, J., Chen, L., & Zhou, B. (1998). Study of porous dielectrics as electret materials. *IEEE transactions on dielectrics and electrical insulation*, 5(1), 58-62.
- [12] Mizsei, J. (2006). Silicon surface passivation by static charge. *Applied surface science*, 252(21), 7691-7699.
- [13] Tschentscher, M., Graber, D., & Franck, C. M. (2020). Influence of humidity on conduction processes in gas-insulated devices. *High Voltage*, 5(2), 143-150.

Reviewer #1 (Remarks to the Author):

The revision made by the authors has strengthened this manuscript. I maintain my recommendation of the first round of review.

ps Figure 2f, the text should be "carbon powder", not "carbon power"

Reviewer #2 (Remarks to the Author):

Authors addressed all the comments raised by the reviewers. This manuscript can be accepted in Nature Communications.

Reviewer #4 (Remarks to the Author):

I will only comment on the response of the authors, i.e., the rebuttal, to reviewer 3. I have not read the manuscript apart from taking a quick look at some of the figures.

First of all, I would like to note that the 6 figures in the main manuscript consist of a huge number of subpanels with tiny labelling. It is not for me to require changes, but it will be difficult for the reader to make sense of the work done.

The following is the list of all the comments made by reviewer 3:

Comment 1:

The reviewer requested a comparison of the author's method to convert potential to charge with finite element analysis simulations (COMSOL or ANSYS). The reviewer would like to see this comparison in terms of uncertainty of the author's method.

Response from authors:

Satisfactory. The authors seemed to have performed COMSOL simulations.

Comment 2:

The reviewer requested an investigation of the origins of charge decay in different materials.

Response from authors:

Satisfactory. Measurements are shown in Fig. 5.

Comment 3:

Why is the visualisation and standardisation of surface charges essential?

Response from authors:

Satisfactory. Reasoning provided.

Comment 4:

"However, the ion deposition process is random and cannot be visualised or standardised. The needle-plate electrode": This is incorrect.

Response from authors:

Satisfactory. Reasoning provided. Literature references given.

Comment 5:

What the authors refer to as standardization is entirely unclear. It appears they are misusing the

word, or have a different meaning intended.

Response from authors:
Satisfactory. The authors changed the wording.

Comment 6:

A large part of the argument is based around triboelectric materials, but all experiments are conducted using corona discharge, which uses a different mechanistic approach to building charge concentration on a surface.

Response from authors:
Satisfactory. I would rather agree with the authors here.

Comment 7:

...it does not say anything about why, how, and the advantages to using AWGCV.

Response from authors:
A bit long and convoluted but satisfactory.

Comment 8:

If the distance between the probe and the sample changes (as it would be the case in Kelvin probe) the transfer matrix H in their algorithm would change. So does this render the algorithm only useful if using exactly the same machine with the same parameters?

Response from authors:
A bit long and convoluted but satisfactory.

Comment 9:

It is unclear what Fig 1.a (first image) is supposed to be communicating. Not covered in the text, but it also just seems like a gimmick rather than a scientific schematic of relevance.

Response from authors:
Somewhat strange but I don't think this cartoon figure harms the message of the work.

Comment 10:

The units and magnitudes in the colour bars are not readable.

Response from authors:

The authors claim to have made them bigger but I still find them incredibly small. This is partly due to the above mentioned, very high number of subpanels.

Comment 11:

The images that portray real data do not have a lateral scale bar.

Response from authors:
Satisfactory.

Comment 12:

Level of scientific English

Response from authors:
Satisfactory.

Dear Reviewers,

We appreciate your efforts in thoroughly reviewing our manuscript entitled “Visualization and Standardized Quantification of Surface Charge Density for Triboelectric Materials” (NCOMMS-23-51061A). We are delighted to notice that this article has been accepted in principle. According to your valuable comments and suggestions, we have made relevant modifications to our manuscript and resubmitted a revised version for consideration. The revisions for addressing the reviewers’ comments have been **highlighted in yellow** in the revised manuscript. Other modifications to discussion and language have been made to improve the manuscript, which has been **highlighted in red** in the revised manuscript. The point-by-point responses to reviewers have been listed as follows.

Thank you very much for your review and consideration of this work.

Sincerely,

Zhong Lin Wang

Professor, Ph.D

Beijing Institute of Nanoenergy and Nanosystems| Chinese Academy of Sciences

Beijing 100083, People’s Republic of China

School of Materials Science and Engineering| Georgia Institute of Technology

Atlanta, Georgia, 30332-0245, United States of America

E-mail: wangzhonglin@binn.cas.cn

Jiaqing Xiong

Professor, Ph.D

Innovation Center for Textile Science and Technology | Donghua University

2999 North Renmin Road, Shanghai 201620, People’s Republic of China

Tel: (86)-19117305006 | E-mail: jqxiong@dhu.edu.cn

Xiaoxing Zhang

Professor, Ph.D

Key Laboratory for High-Efficiency Utilization of Solar Energy and Operation Control of Energy Storage System| School of Electrical and Electronic Engineering| Hubei University of Technology

Nanli Road, Wuhan 430068, People’s Republic of China

State Key Laboratory of Power Grid Environmental Protection| School of Electrical Engineering and Automation| Wuhan University

Bayi Road, Wuhan, Hubei 430072, People's Republic of China

Tel: (86)-13627275072 | E-mail: zhangxx@hbut.edu.cn

Tao Shao

Professor, Ph.D

Beijing International S&T Cooperation Base for Plasma Science and Energy Conversion| Institute of Electrical Engineering| Chinese Academy of Sciences

No. 6, North Ertiao, Zhongguancun, Beijing 100190, People's Republic of China

Tel: (86)-13810312461 | E-mail: st@mail.iee.ac.cn

Point-to-point responses

The revisions in response to the Reviewers' comments are highlighted in yellow in the revised manuscript.

We have also made careful revisions to improve the manuscript. All the relevant changes are marked in red in the revised manuscript.

Author's Response to Reviewer 1

Reviewer #1: The revision made by the authors has strengthened this manuscript. I maintain my recommendation of the first round of review.

ps Figure 2f, the text should be "carbon powder", not "carbon power"

Response: We highly appreciate your positive feedback and diligent review of this work. We have corrected the type error from "carbon power" to "carbon powder" in the revised version. We also checked the whole manuscript carefully and corrected relevant errors. Thanks again for your approval and consideration.

Author's Response to Reviewer 2

Authors addressed all the comments raised by the reviewers. This manuscript can be accepted in Nature Communications.

Response: We highly appreciate your positive feedback and diligent review of this work. We have carefully checked and revised the whole manuscript based on the comments of reviewers and editors. Thanks again for your approval and consideration.

Author's response to Reviewer 4

I will only comment on the response of the authors, i.e., the rebuttal, to reviewer 3. I have not read the manuscript apart from taking a quick look at some of the figures.

First of all, I would like to note that the 6 figures in the main manuscript consist of a huge number of subpanels with tiny labelling. It is not for me to require changes, but it will be difficult for the reader to make sense of the work done.

Response: Thanks for your valuable comments and careful review of this work. We have redrawn Figure 1-Figure 6 in the revised manuscript to ensure the subpanels with tiny labeling are legible. Thanks again for your suggestions.

The following is the list of all the comments made by reviewer 3:

Comment 1:

The reviewer requested a comparison of the author's method to convert potential to charge with finite element analysis simulations (COMSOL or ANSYS). The reviewer would like to see this comparison in terms of uncertainty of the author's method.

Response from authors:

Satisfactory. The authors seemed to have performed COMSOL simulations.

Response: Thanks for your satisfaction with our response.

Comment 2:

The reviewer requested an investigation of the origins of charge decay in different materials.

Response from authors:

Satisfactory. Measurements are shown in Fig. 5.

Response: Thanks for your satisfaction with our response.

Comment 3:

Why is the visualisation and standardisation of surface charges essential?

Response from authors:

Satisfactory. Reasoning provided.

Response: Thanks for your satisfaction with our response.

Comment 4:

"However, the ion deposition process is random and cannot be visualised or standardised. The needle-plate electrode": This is incorrect.

Response from authors:

Satisfactory. Reasoning provided. Literature references given.

Response: Thanks for your satisfaction with our response.

Comment 5:

What the authors refer to as standardization is entirely unclear. It appears they are misusing the word, or have a different meaning intended.

Response from authors:

Satisfactory. The authors changed the wording.

Response: Thanks for your satisfaction with our response.

Comment 6:

A large part of the argument is based around triboelectric materials, but all experiments are conducted using corona discharge, which uses a different mechanistic approach to building charge concentration on a surface.

Response from authors:

Satisfactory. I would rather agree with the authors here.

Response: Thanks for your satisfaction with our response.

Comment 7:

...it does not say anything about why, how, and the advantages to using AWGCV.

Response from authors:

A bit long and convoluted but satisfactory.

Response: Thanks for your comments. Considering that this concern involves algorithms, we provided comprehensive explanations in our response. Thanks again for your approval and satisfaction.

Comment 8:

If the distance between the probe and the sample changes (as it would be the case in Kelvin probe) the transfer matrix H in their algorithm would change. So does this render the algorithm only useful if using exactly the same machine with the same parameters?

Response from authors:

A bit long and convoluted but satisfactory.

Response: Thanks for your comments. Considering that this concern involves algorithms, we provided comprehensive explanations in our original response.

Comment 9:

It is unclear what Fig 1.a (first image) is supposed to be communicating. Not covered in the text, but it also just seems like a gimmick rather than a scientific schematic of relevance.

Response from authors:

Somewhat strange but I don't think this cartoon figure harms the message of the work.

Response: Thanks again for your approval and satisfaction with our response.

Comment 10:

The units and magnitudes in the colour bars are not readable.

Response from authors:

The authors claim to have made them bigger but I still find them incredibly small. This is partly due to the above-mentioned very high number of subpanels.

Response: Thanks for your comments. We further modified the layout of the Figures in the manuscript and increased the font size of subpanels to ensure legibility. Thanks again for your suggestions.

Comment 11:

The images that portray real data do not have a lateral scale bar.

Response from authors:

Satisfactory.

Response: Thanks again for your approval and satisfaction with our response.

Comment 12:

Level of scientific English

Response from authors:

Satisfactory.

Response: Thanks again for your approval and satisfaction with our response.